# Optimization of wind farm portfolio for minimizing overall power fluctuations at selective frequencies - a case study of the Faroe Islands

Turið Poulsen[1], Bárður A. Niclasen[1], Gregor Giebel[2], and Hans Georg Beyer[1]

[1]Faculty of Science and Technology, University of the Faroe Islands
[2]Department of Wind Energy, Technical University of Denmark

**Correspondence:** Turið Poulsen (turidp@setur.fo)

**Abstract.** Hourly modeled wind turbine power output time series - modeled based on outputs from the mesoscale numerical weather prediction system WRF - are used to examine the spatial smoothing of various wind farm portfolios located on a complex isolated island group with a surface area of 1400 km$^2$. Power spectral densities (PSD), hourly step-change functions, and duration curves are generated, and the 5th and 95th percentiles and the standard deviations of the hourly step-change functions

are calculated. The spatial smoothing is identified from smaller high-frequency PSD amplitudes, lower hourly fluctuations, and more flat duration curves per installed wind power capacity, compared with single wind turbine outputs. A discussion on the limitation of the spatial smoothing for the region is included, where a smoothing effect is observed for periods of up to 1-2 days, although most evident at higher frequencies. By maximizing the smoothing effect, optimal wind farm portfolios are presented with the intention of minimizing overall wind power fluctuations. The focus is mainly on the smoothing effect on

the 1-3 hourly time scale, during which the coherency between wind farm power outputs is expected to be dependent on how the regional weather travels between local sites, thereby making optimizations of wind farm portfolios relevant; in oppose to a focus on either lower or higher frequencies on the scale of days or minutes, respectively, during which wind farm power output time series are expected to be either close to fully coherent due to the same weather conditions covering a small region or not coherent as the turbulences in separate wind farm locations are expected to be uncorrelated. Results show that an optimization

of the wind farm capacities at fourteen pre-defined wind farm site locations has a minimal improvement on the hourly fluctuations compared with a portfolio with equally weighted wind farm capacities. However, choosing optimized combinations of individual wind farm site locations decreases the 1-3 hourly fluctuations considerably. For example, selecting a portfolio with four wind farms (out of the fourteen pre-defined wind farm site locations) results in 15% lower 5th and 95th percentiles of the hourly step-change function when choosing optimal wind farm combinations compared with choosing the worst wind farm

combinations. For an optimized wind farm portfolio of seven wind farms, this number is 13%. Optimized wind farm portfolios consist of distant wind farms, while the worst portfolios consist of clustered wind farms.

# 1 Introduction

Most nations strive to make their electricity generation less dependent on fossil fuels. Wind power is an attractive solution, being environmentally friendly, mature, and affordable. However, an increase in the wind power contribution to the power grid can cause challenges as wind varies across various time scales. Measures are required to mitigate natural wind power fluctuations in order to balance and keep the grid stabilized. One possible measure is spatial distribution of wind farm siting, see e.g. Beyer et al. (1990) or Katzenstein et al. (2010). Isolated grids face special challenges, as they usually span across small areas. Therefore, the available wind resource is partly or fully co-varying, which poses a serious challenge for maintaining the energy balance of the grid.

The wind power fluctuations in a confined isolated region are examined in this study, in which the Faroe Islands are used as a case study. The Faroe Islands are a small isolated mountainous island group (1400 km$^2$) in the north-east of the Atlantic Ocean, ~300 km away from any mainland, see Fig. 1. The topography is complex, and the climate is windy. Battery systems have been installed in the Faroe Islands and can handle wind variability up to timescales of several minutes. Manual set-point-control of hydro- and fossil power plants is presently used to compensate for wind variability on timescales from several minutes to several days. In the future, the contribution from wind power will increase rapidly, and the present manual-control-regime will be put under pressure. There is therefore considerable local interest in limiting the inherent wind-variability on hourly timescales in the Faroe Islands.

The aim of the study is to analyze and optimize the smoothing effect caused by the nature of dispersed wind farm sites in the region. Wind farm portfolios are optimized with the objective of minimizing the 2-3 hourly power output fluctuations.

There are numerous studies in the literature on analyzing the smoothing effect of spatially distributed wind farm sites on various time and spatial scales, applying different techniques, e.g. correlation analysis (Beyer et al., 1993; Katzenstein et al., 2010; Pearre and Swan, 2018; Giebel, 2000), step-change analysis (Beyer et al., 1990, 1993; Katzenstein et al., 2010; Pearre and Swan, 2018; Giebel, 2000), visualization of duration curves (Giebel, 2000; Barasa and Aganda, 2016; Katzenstein et al., 2010), and analysis in the frequency domain (Beyer et al., 1990, 1993; Nanahara et al., 2004b, a; Katzenstein et al., 2010; Poulsen and Beyer, 2020). One advantage of frequency analysis is the ability to characterize the extent of the smoothing effect for many frequencies. The reduction in the total variance of aggregated wind farm data can be small even though a significant smoothing effect occurs at higher frequencies, as observed by Beyer et al. (1990). In contrast, time domain analysis - such as step-change analysis and duration curves - may be more intuitive to interpret. Other examples of studies analyzing the smoothing of spatially distributed wind farms in the time domain include Palutikof et al. (1990), Wan et al. (2003), and Giebel (2001).

Frank et al. (2021) is an example of a recent study examining the smoothing effect of the natural variability in both wind and solar power. They analyzed daily data from eleven European countries. Although their focus was both on wind and solar power, the smoothing effect of spatially distributed wind power can be seen in their results, based on the low correlation coefficients and low joint probabilities of the minimum and maximum wind power between countries. In addition, they optimize the installation ratio between solar and wind power in order to reduce extreme fluctuations. Even though their results cannot be

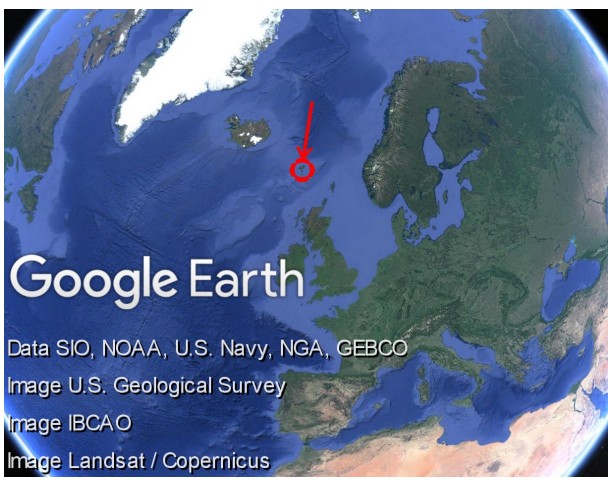

**Figure 1.** Map generated with the software © Google Earth Pro (see affiliation on the map). The red circle marks the position of the Faroe Islands.

directly compared with the results in this study, their study demonstrates the potential of balancing the natural fluctuations from renewable energy resources by considering the configuration of the resources.

Reichenberg et al. (2014) and Cassola et al. (2008) are examples of studies focusing on minimizing fluctuations in the total wind power production through wind farm optimization. Reichenberg et al. (2014) developed a method for optimizing the

locations of wind farms by minimizing the coefficient of variation of the total wind power output time series. Their method is based on sequential optimization of site localization, and it is applied in the Nordic countries and Germany. Cassola et al. (2008) proposed a procedure for optimally distributing the relative size of considered wind farms by minimizing either the variation in the total wind power output time series, or by minimizing the ratio between the variation and the total wind energy production. Their method was tested with regards to wind power data at ten locations in Corsica (France), a complex island in

the Mediterranean Basin stretching 175 km in the latitude direction and 80 km in the longitude direction with a mountain chain crossing the islands in the north-south direction and peaks higher than 2000 m. However, the results from these studies cannot be directly compared to the results obtained in this study, as the methodologies are different as well as the geography.

The results in the present study are based on hourly modeled wind turbine power output data, which are analyzed by generating power spectral densities (PSD), hourly step-change functions, and duration curves for individual and lumped power

output time series. Moreover, the 5th and 95th percentiles and the standard deviations of the hourly step-change functions are extracted for quantitatively comparing the results. The PSDs are further used to optimize wind farm portfolios by identifying the wind farm combinations with the lowest 2-3 hourly wind power fluctuations.

The methodology applied in the presented wind farm portfolio optimization study is unconventional - using spectral results as the objective function for minimizing the overall wind power fluctuations, a powerful methodology in the sense that the

optimization focuses solely on the preferred frequencies. No such optimization study has ever been conducted for the Faroe Islands region.

The datasets used in this study are introduced in Sect. 2, the methods applied are described in Sect. 3, the results are presented in Sect. 4, and a summary and conclusions are given in Sect. 5.

## 1.1 A note on ignoring the wind farm smoothing effect - Paper assumption

In this study, turbine outputs are analyzed, thus neglecting the potential smoothing effects within wind farms. Various studies investigate the coherence function between time series on a wind farm scale (Schlez and Infield, 1998; Vigueras-Rodríguez et al., 2012; Vincent et al., 2013); the coherence function can be used as a measure of the smoothing effect between time series as a function of frequency. If the coherence is high, the smoothing effect is small in the considered frequency range. If the coherence is small, a smoothing effect occurs.

The distance between the two farthest turbines in the largest wind farm in the Faroe Islands at the time of the preparation of this manuscript - with thirteen 0.9 MW turbines - is about 670 m. For a wind speed of $10 \, \text{m s}^{-1}$ (approximate average values as observed for the Faroe Islands region by Poulsen et al. (2021)), a distance of 670 m, and a frequency of $(2 \, \text{h})^{-1}$, the analytical coherence model presented by Vigueras-Rodríguez et al. (2012) yields squared coherence values of 0.81 and 0.92 in the lateral and longitudinal directions, respectively, and the wind farm smoothing effect is therefore expected to be small for frequencies of up to $(2 \, \text{h})^{-1}$. For larger wind farms, a wind farm smoothing effect is expected. Although this is outside the scope of this study, a more detailed study could apply methods such as those presented by Nørgaard and Holttinen (2004) or Sørensen et al. (2008) when scaling up turbine outputs to large wind farms.

## 2 Data

Hourly WRF-generated wind speed data is the primary dataset used throughout this study. Time series are extracted at locations with favorable wind farm conditions. In addition, available empirical data are used for comparison.

## 2.1 WRF-generated wind speed data

WRF-generated data were simulated by Kjeller Vindteknikk and made available to us by SEV (the local power company of the Faroe Islands). The model setup is documented in Haslerud (2019). Multiple output parameters were generated, including the wind speed data used in this study, validated by Poulsen et al. (2021). WRF version v3.8.1 was used for the simulations, with 51 vertical levels (8 in the lower 200 m). Three nested domains were used, with horizontal resolutions of 500 m x 500 m, 1500 m x 1500 m, and 7500 m x 7500 m, for the innermost, middle, and outer domains, respectively. The innermost domain covered the entire Faroe Islands. Data was stored every hour during the period from 1. July 2016 to 30. June 2018. The Thompson scheme, the Mellor-Yamada-Janjik scheme, and the NOAH scheme were applied for microphysics, boundary layer mixing, and the surface, respectively. Every 3 hours, ERA5 reanalysis data with a horizontal resolution of approximately 0.25 degrees was used as boundary condition (available from the European Centre for Medium-Range Weather Forecasts, ECMWF).

## 2.2 Empirical data

During the period from 1. July 2016 to 30. June 2018, hourly power production data from three wind farms operating on the Faroe Islands are available. The hub heights of all turbines (except one smaller) are 45 m a.g.l. In addition, 10 min averaged wind speed measurements from an additional site at a height of 52.8 m a.g.l. during the period from 21. July 2016 to 30. June 2018 are available. The locations of the four sites are pinpointed in Fig. 2.

## 2.3 Favorable wind farm site locations

Magnussen (2017) presented a map with current and potential wind farm site areas on the Faroe Islands, see Fig. 2. The areas were selected based on modeled wind resource, distance from the road, distance from the high voltage grid, terrain slope, populated areas, and presence of water and rivers. These considerations were weighted and merged. Based on additional considerations, as e.g. expected turbulent areas, some locations were rejected, and a selection of favorable wind farm sites was composed.

Data at these favorable locations are used throughout the study.

## 3 Method

Modeled wind turbine power output data at a height of 45 m a.g.l from the period of July 2016 to June 2018 are examined. The data processing of the time series is explained in the following subsection. The applied analytical methods are defined in the second and third subsections.

## 3.1 Data processing

WRF-generated wind speed time series for selected sites are vertically interpolated to a height of 45 m a.g.l. This height is chosen because all operating wind turbines in the Faroe Islands at the time of the preparation of this manuscript had a hub height of 45 m a.g.l. The interpolated time series are subsequently corrected for the aliasing effect, as elaborated on in the appendix.

The wind speed time series are modeled to power output time series using the power curve of an Enercon E-44 wind turbine with a storm control function (Enercon, 2012) and a rated power of 0.9 MW. This turbine model is chosen because most of the currently operating wind turbines in the Faroe Islands are of the type Enercon E-44.

To be able to compare results, all power output time series are normalized by their rated wind power capacity. When time series are aggregated, the lumped time series is computed prior to normalization.

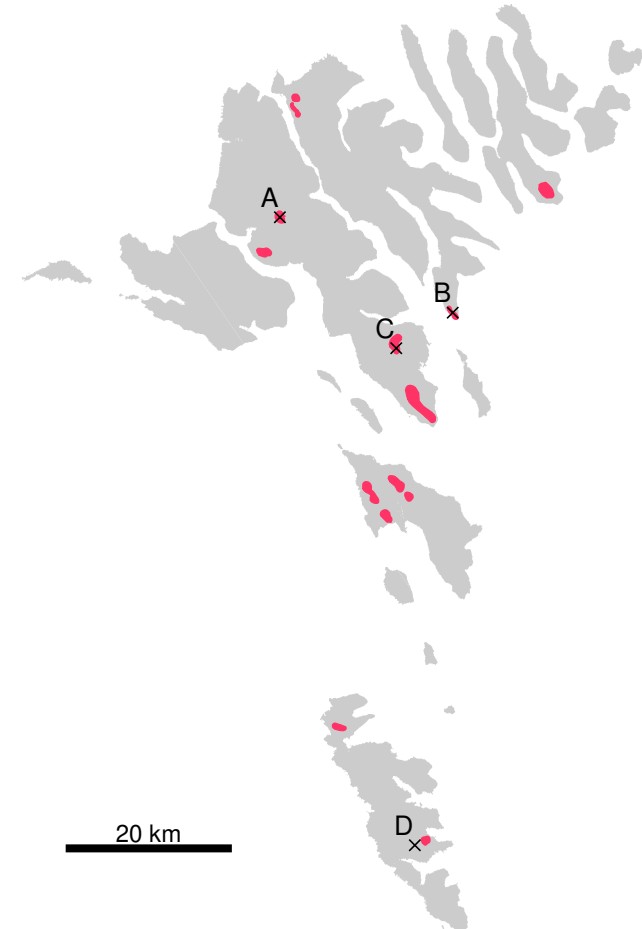

**Figure 2.** Gray areas mark the terrain of the Faroe Islands. Red patches mark current and potential wind farm locations as selected by Magnussen (2017). Black markers pinpoint sites from which actual measured data are available for the period of June 2016 to July 2018, denoted by the letters A to D. The area of the terrain was created using a 10 m raster map extracted from https://www.foroyakort.fo/ on 18. December 2020, created in Denmark from satellite data from 2017.

### 3.2 Approaches for the characterization of the power output time series

In this study, three analysing techniques are applied to the time series - spectral analysis, step-change analysis, and the generation of duration curves - each of which is elaborated on in the following subsections. The first two characterize the time series
135 fluctuations, while the latter characterizes the distribution of the power generation.

### 3.2.1 Spectral analysis

PSDs are generated for normalized power output time series using discrete Fourier transform. Only the amplitudes corresponding to the positive frequencies of the discrete Fourier transform are extracted. Thus, the integral of the raw spectra with respect to frequency yields about half of the variance of the time series. Due to the inherent uncertainty of raw spectral estimates, smoothed PSDs are generated by dividing the time series into chunks, calculating the spectral results for each chunk, then averaging over all chunks. For each chunk, the average value is subtracted, and a hamming window is applied. A 50% overlap between chunks is used.

The length of the chunks is a compromise between the accuracy of the PSD estimates (smaller chunks, i.e., more chunks) and the frequency resolution and the lowest resolvable frequency (longer chunks). In this study, a length of 256 data points was chosen (10 days and 16 hours), giving 135 overlapping chunks for the two-year long hourly time series. The PSD estimates will therefore be generated for frequencies between $(256 \text{ h})^{-1}$ (thus, including PSD estimates for the 3-4 day period of the time scale of migratory low-pressure systems at mid and high latitudes) and the Nyquist frequency of $(2 \text{ h})^{-1}$ with a resolution of $(256 \text{ h})^{-1}$.

### 3.2.2 Step-change analysis

The step-change function of the wind power time series is calculated as given in Eq. (1).

$$\Delta P_t = P_{t+1} - P_t \tag{1}$$

where $P_t$ is the wind power production at time $t$, $P_{t+1}$ is the wind power production of the consecutive time step, and $\Delta P_t$ is the corresponding change in the wind power production between these two time steps. The distribution of the wind power step-change function is presented with its probability density function (pdf). In addition, the 5th and 95th percentiles and the standard deviation of the step-change function are identified. Although the hourly step-change function is not normally distributed, the standard deviation is a measure for characterizing the hourly fluctuations of the time series. The higher the standard deviation, the more frequent large hourly fluctuations occur, and vice versa. The 5th and 95th percentiles are measures characterizing the more extreme hourly fluctuations, whereby the hourly fluctuations are beyond either of these values ten percent of the time.

### 3.2.3 Extraction of power duration curves

The power output time series data are sorted according to descending magnitude to extract the fraction of exceedance of the power levels, presenting the power duration curve.

### 3.3 Optimization method

By optimizing wind farm portfolios with the aim of minimizing the fluctuations of the total time series, the most stable wind farm configurations are derived. For this purpose, modeled wind turbine power output time series are used, positioned at the

favorable wind farm site locations shown in Fig. 2, and modeled as described in Sect. 3.1. The overall power output time series of the wind farm portfolio, $P_{lumped}$, is generated by aggregating the turbine power output time series of the combination of wind farms in the given portfolio, as demonstrated in Eq. (2).

$$P_{lumped}(t) = \sum_{i=1}^{n} P_{inst,i} P_i(t) \tag{2}$$

where $P_i(t)$ is the modeled wind turbine power output time series at location $i$, and $n$ is the number of considered wind farm site locations in the given portfolio. $P_{inst,i}$ is the relative wind farm capacity at location $i$ with respect to the total capacity of the wind farm portfolio. The relative capacity is used in order to normalize the aggregated time series, so it represents wind power output data per installed wind power capacity, as mentioned in Sect. 3.1. Thus, by definition, the sum of $P_{inst,i}$ must equal one:

$$\sum_{i=1}^{n} P_{inst,i} = 1 \tag{3}$$

where $P_{inst,i}$ is always greater or equal to zero:

$$\forall i \leq n, \qquad P_{inst,i} \geq 0 \tag{4}$$

As PSD estimates represent time series fluctuations with respect to frequency, the *objective function* in the optimization is set to minimize the integral of the PSD estimates of the aggregated power output time series from the wind farm portfolio (Eq. (2)) for frequencies between $(2\,\mathrm{h})^{-1}$ and $(3\,\mathrm{h})^{-1}$:

$$\text{minimize} \quad \int_{(3\mathrm{h})^{-1}}^{(2\mathrm{h})^{-1}} \text{PSD d}f \tag{5}$$

with the constraints given in Eq. (3) and Eq. (4). Thus, the optimization minimizes the two-three hourly fluctuations of the wind farm portfolio power output time series. These are the highest resolvable spectral frequencies when working with hourly data. Lower frequencies could also be considered, but are not chosen, because larger periods, on the scale of days, are expected to be close to fully correlated across the Faroe Islands, as the same weather systems are expected to cover the small island region ($1400\,\mathrm{km}^2$). This is discussed further in Sect. 4.3 and Sect. 4.5. The *optimized parameters* are set to be one of the two:

1) the normalized wind power capacities, $P_{inst,i}$, where the locations of the wind farms in the portfolio, $i$, are fixed.

2) the locations of the wind farms, $i$, where the total number of wind farms in the portfolio, $n$, is fixed, and the normalized wind power capacities are equal at all site locations ($1/n$).

Optimized portfolios using the first mentioned parameters are presented in Sect. 4.2, while optimized portfolios using the latter mentioned parameters are presented in Sect. 4.4.

## 4 Results

All results represent normalized power output time series with respect to the total rated power. This enables comparisons between results.

First, statistical characteristics of modeled turbine power output time series from WRF-generated wind speeds are compared with those of measured data. In the subsequent subsection, the modeled turbine power output time series are used to optimize wind farm capacities at selective wind farm site locations, with the objective of minimizing the high frequency fluctuations of the aggregated time series. Limitations of the spatial wind farm smoothing effect in the small island system are discussed in the third subsection. An optimization of the geographical distribution of wind farms is presented in the fourth subsection. The last subsection includes a discussion on the potential optimization effects for minimizing the overall power output time series for a variety of frequency ranges.

### 4.1 Comparing statistical characteristics of WRF modeled and measured dataset

To examine the validity of the modeled turbine power output data based on WRF-generated wind speeds, these time series are compared to actual measurements. Three wind farms were operating on the Faroe Islands during the period of July 2016 to June 2018, the period of the WRF-generated data. One additional measured time series is gained by converting meteorological data into turbine output data.

PSD, pdf of hourly step-change functions, and duration curves for site-specific and lumped wind farm power output data are calculated and depicted in Fig. 3; the lumped time series weights the installed capacity at the four site locations equally. The site-specific PSD and pdf display similar behaviors as observed in the measurements. However, the PSD amplitudes at the highest frequencies are somewhat overestimated at sites B and C. This is also clear from the 5th and 95th percentiles and the standard deviations of the hourly step-change functions presented in Table 1. The duration curves have evident site-specific deviations.

**Table 1.** Standard deviation ($\sigma$) and the 5th and 95th percentiles of the hourly step-change function of the power outputs modeled based on WRF-generated wind speeds ($\Delta P_{WRF}$) and empirical data ($\Delta P_{Emp}$) (see Fig. 2 for site locations). The lumped time series assumes an equal distribution of the wind power capacity over all four locations. All time series are normalized by their rated power.

|  |  | site A | site B | site C | site D | lumped time series |
|---|---|---|---|---|---|---|
| | $\sigma$ | 0.124 | 0.127 | 0.113 | 0.108 | 0.071 |
| $\Delta P_{WRF}$ | 5th percentile | -0.200 | -0.200 | -0.183 | -0.172 | -0.113 |
| | 95th percentile | 0.202 | 0.201 | 0.174 | 0.170 | 0.117 |
| | $\sigma$ | 0.122 | 0.115 | 0.097 | 0.109 | 0.069 |
| $\Delta P_{Emp}$ | 5th percentile | -0.194 | -0.190 | -0.156 | -0.180 | -0.110 |
| | 95th percentile | 0.199 | 0.188 | 0.153 | 0.180 | 0.114 |

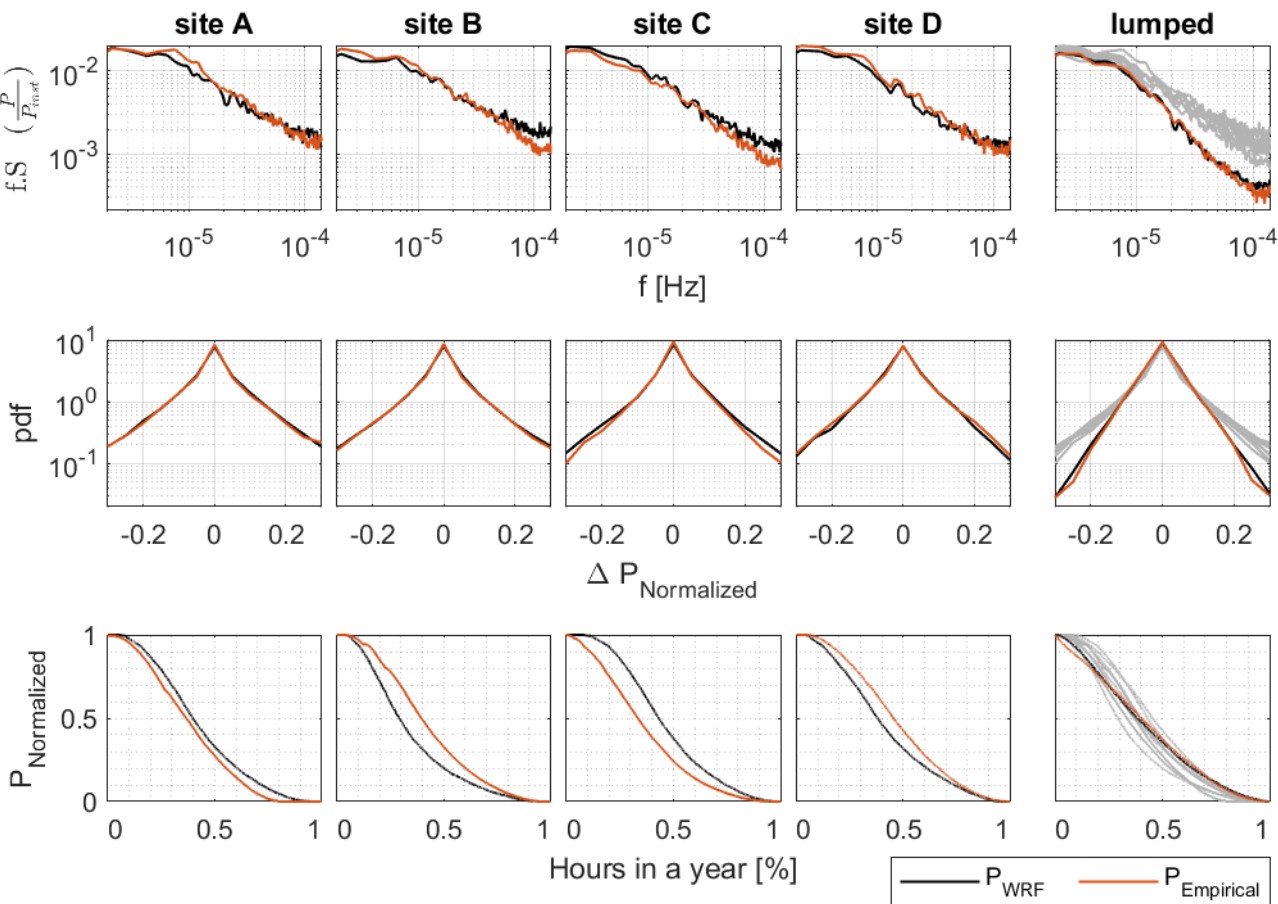

**Figure 3.** PSD (top), pdf of hourly step-change functions (middle), and duration curves (bottom) of hourly wind power output time series per installed capacity ($\frac{P}{P_{inst}}$). Black colors represent results developed from WRF-generated data. Red colors represent results developed from measured data. The first four columns represent data from individual sites. The last column represents the lumped time series of all four sites, with an equal distribution of the wind power capacity over all four locations.

## 4.2 Optimization of wind farm capacities

By optimizing the wind power capacity of scattered wind farms with the aim of minimizing the fluctuations of the total time series, the most stable wind farm configuration is derived. For this purpose, modeled wind turbine output data from predefined locations are used - the favorable wind farm locations in the region as selected by Magnussen (2017), see Sect. 2.3. The objective of the optimization is to minimize the 2-3 hourly fluctuations, see Sect. 3.3. The optimization algorithm is conducted for three cases:

Case I:     Considering all fourteen favorable wind farm site locations mapped in Fig. 2.

Case II:    Excluding the two southernmost wind farm site locations.

Case III:   Upper and lower boundaries for the installed wind power capacity are given for each of the fourteen wind farms.[1]

The optimized wind farm configurations for the three cases are presented in Fig. 4. The distributions of the optimized wind farm capacities are logical, with generally more capacities at more remote sites and less capacities for closely clustered wind farm sites. However, the effect of the optimization is limited compared with if the wind power capacities were to be equally distributed over all considered sites. The reduction in the 5th and 95th percentiles and standard deviations of the hourly step-
change functions are 2% or less for the three optimized cases. It may be speculated that the observed limited hourly smoothing effect is because the improvement that the optimization adds is small compared with the smoothing that already occurred by considering the equally weighted spatial distribution of the wind farms.

To investigate this further, small wind farm portfolios in the region are examined by looking at combinations of three wind farms. For the combination of the southernmost site and the two clustered northernmost sites, optimized wind farm capacity
distribution places around 27% of the total power capacity of the wind farm portfolio at each of the northernmost sites and almost 47% at the southernmost site. The distribution is logical, with lower capacities at both of the clustered wind farm sites, and higher capacity at the distant site. If the two clustered sites were fully correlated on a 2-3 hourly scale while the distant site was uncorrelated, the capacity distribution would be expected to be 25%, 25%, and 50%, respectively, which is close to what is observed for this set-up. For the optimized portfolio with this set-up, the 5th and 95th percentiles and the standard
deviation of the hourly power output step-change function are 4-5% lower compared with if the wind power capacity were to be equally distributed over the three sites. Comparing the optimized wind farm portfolio to a portfolio in which 40% of the total wind power capacity is placed at each of the two northernmost wind farms and 20% at the southernmost wind farm – a rather illogical distribution in terms of spatial smoothing of the power capacity – the optimized wind farm portfolio contains 12-13% lower 5th and 95th percentiles and standard deviation. This indicates that the power capacity distribution of the wind
farms matters.

Going back to case I (all fourteen favorable wind farm sites in the region), although the optimized portfolio has only limited effect on the smoothing effect compared with the portfolio in which the power capacities are equally distributed, comparing the overall power outputs of the optimized distribution of case I to the power output of the equally distributed capacity of case III with the constraint of the given boundary conditions, the 5th and 95th percentiles and the standard deviation of the hourly
step-change function are 4% lower. Likewise, by comparing the power output of the optimized distribution of case I to a set-up where 11% of the total wind power capacity is placed at the four closest clustered sites as well as at the two northernmost sites,

---

[1]Upper bounds are defined based on the site-specific areas as marked in Magnussen (2017). For every 30000 $m^2$ ($3 \cdot 44$ m $\cdot 5 \cdot 44$ m $\sim$30000 $m^2$), 0.9 MW is added to the possible upper boundary. The rotor diameter of an Enercon E-44 turbine is 44 m, and it currently is the most frequently used turbine model on the Faroe Islands. However, none of the optimized values exceed the upper boundaries, making the upper boundaries redundant. The lower boundaries are set to the capacity of the current and committed wind farms on the Faroe Islands as distributed in the study by Tróndheim et al. (2021), with a total of 60.63 MW. Values for both the upper and lower boundaries are normalized by a total power capacity of 168 MW, being the proposed wind power capacity in 2030 by Tróndheim et al. (2021).

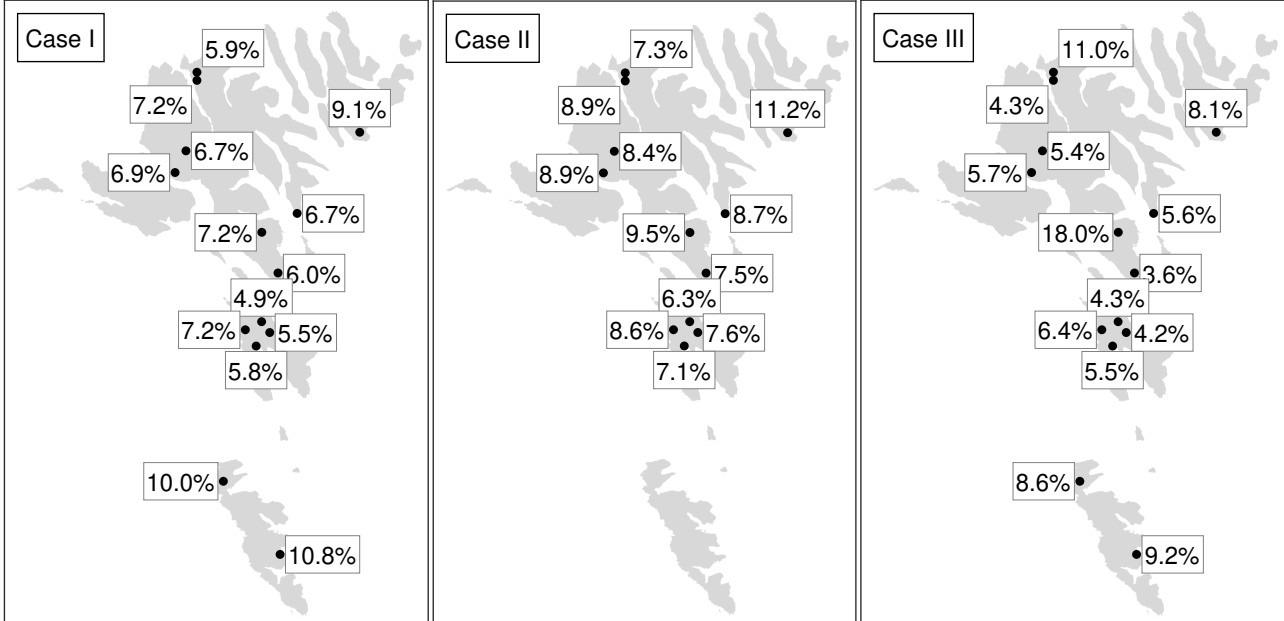

**Figure 4.** Optimized wind farm configurations, shown in percentages of site-specific wind power capacity per total installed wind power capacity. Case I: Considering all fourteen favorable wind farm locations selected by Magnussen (2017). Case II: excluding the two southernmost sites. Case III: setting site-specific lower and upper boundaries for the wind farm capacities at the fourteen sites. The maps of the terrain are created using a 10 m raster map extracted from https://www.foroyakort.fo/ on 18. December 2020, created in Denmark from satellite data from 2017.

and 4.5% at the rest of the sites (with the intent to distribute the capacity of the wind farms poorly), the 5th and 95th percentiles and the standard deviation of the hourly step-change function are 6% and 8% lower, respectively. Again, this indicates that the wind power capacity distribution matters.

It is expected that the results are different for other case studies. The magnitude of the smoothing due to the spatial distribution of the wind farm capacities depends on the magnitude of the coherences and the details of the shape of the PSD curves. Other locations with e.g. simpler terrain are expected to have larger coherencies. In contrast, larger regions have the advantage of being able to have scattered wind farms to a larger extent.

### 4.2.1 Comparison of optimized portfolios according to different conditions

PSD, pdf of the hourly step-change functions, and the duration curves of the optimized lumped time series are displayed in Fig. 5. For reference, results from single turbine outputs are superimposed. The 5th and the 95th percentiles and the standard deviations of the step-change functions are given in Table 2.

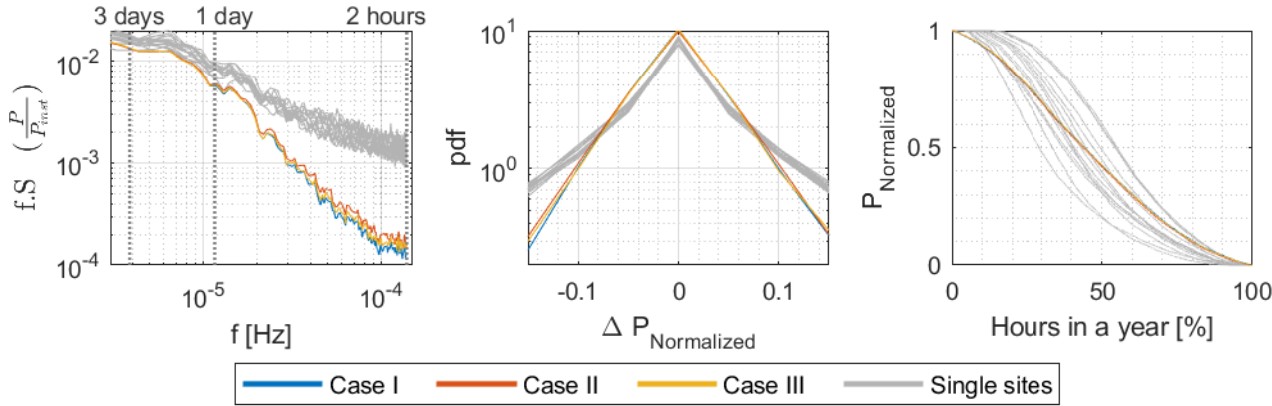

**Figure 5.** PSD (left), pdf of hourly step-change functions (middle), and duration curves (right) of lumped hourly wind power output time series per installed capacity $(\frac{P}{P_{inst}})$ for the three optimized cases displayed in Fig. 4, as well as for each single turbine output time series (gray).

**Table 2.** Standard deviation ($\sigma$) and the 5th and the 95th percentiles of the hourly step-change functions per rated wind power capacity for the three optimized lumped power output time series displayed in Fig. 4.

| percentiles | case I | case II | case III |
|---|---|---|---|
| $\sigma$ | 0.0558 | 0.0599 | 0.0574 |
| 5th percentiles | -0.0892 | -0.0954 | -0.0917 |
| 95th percentiles | 0.0900 | 0.0964 | 0.0923 |

No clear distinction can be observed in the pdf of the step-change functions and the duration curves of the three cases. However, a smoothing effect is observed compared with single turbine outputs with lower hourly step-changes and less frequent intervals with zero and rated power production.

The spectra for all cases are equivalent for low frequencies and similar to those of the single turbine outputs, while the energy content at higher frequencies differs. A smoothing effect for the combined time series (cases I-III) compared with the single turbine outputs is most evident at the highest frequencies, but noticeable for periods of up to 1-2 days.

Out of the three cases, case II (the exclusion of the two southernmost site locations) has the heaviest spectral tail, followed by case III (bound by current and committed wind farm power capacities as distributed in the study by Tróndheim et al. (2021)) and finally case I (all site locations). The 5th and 95th percentiles and the standard deviations of the hourly step-change functions of the time series show the same trend, with the highest values for case II, followed by case III, then case I. Excluding the two southernmost site locations increases the 5th and 95th percentiles and the standard deviation of the hourly step-change function by 7%, while setting boundaries to the lower and upper wind farm capacities increases the percentiles and standard deviation by 3%.

#### 4.2.2 Sensitivity analysis on the PSD of power outputs with the inclusion of remote sites

As observed in Fig. 5 and Table 2, excluding the two time series of the southernmost island increases the optimized lumped time series fluctuation for the highest frequency range considerably. In other words, the inclusion of the two southernmost site locations smoothes the total time series.

To test if the smoothing effect originates from the inclusion of two additional sites in the given region, or whether it is because the two sites are further away, PSD for various optimized site combinations are generated and displayed in Fig. 6:

- Including all fourteen sites (case I)

- Excluding the two southernmost sites (case II)

- All possible combinations of twelve sites with the constraint that two out of the twelve sites are those located on the southernmost islands

Case II has higher PSD amplitudes in the highest frequency range compared with other combinations of the twelve site locations that include the two southernmost site locations. This indicates that the smoothing effect for the highest frequencies is more pronounced for remote sites.

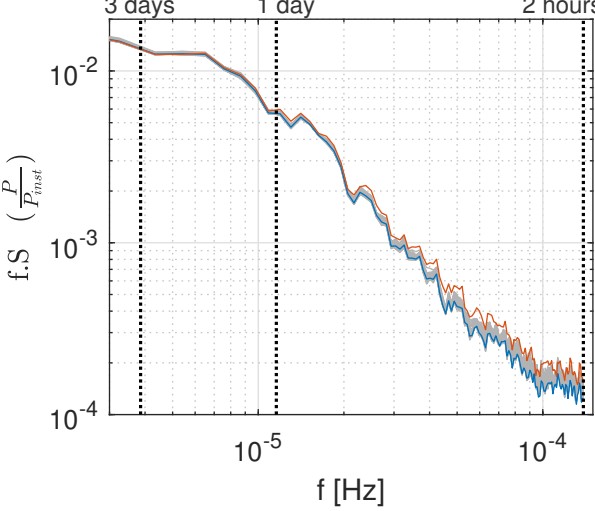

**Figure 6.** PSD for various optimized combinations of lumped hourly wind power output time series per installed capacity ($\frac{P}{P_{inst}}$). Blue line: case I. Red line: case II. Gray lines (sixty-six compressed lines): PSD for all sixty-six possible combinations of twelve lumped power output time series, with the constraint that two of the twelve sites are located on the southernmost islands.

### 4.3 Limitation of spatially distributed wind farms

To test the limitations of the smoothing effect from spatially distributed wind farms in the Faroe Islands, PSDs are generated for multiple combinations of up to fourteen turbine power output time series; the fourteen favorable site locations marked in Fig. 2.

Each computation considers an equal distribution of the rated power over all considered sites. Results are displayed in Fig. 7. As the number of lumped time series, $n$, increases, the PSD amplitudes at the highest frequencies decrease. The smoothing effect is evident for periods of up to 1-2 days, but more evident for the higher frequencies. The smoothing effect observed when adding one additional wind farm to the lumped time series becomes less pronounced when $n$ is large. These characteristics are similar to results observed by e.g. Katzenstein et al. (2010) and Beyer et al. (1993), who analyzed the variability of interconnected wind power time series spatially dispersed in the area of Texas and northwest Germany, respectively.

The PSD of the total power output when combining $n$ wind farms differ for the various combinations of the individual wind farms. This will be analyzed further in the following subsection.

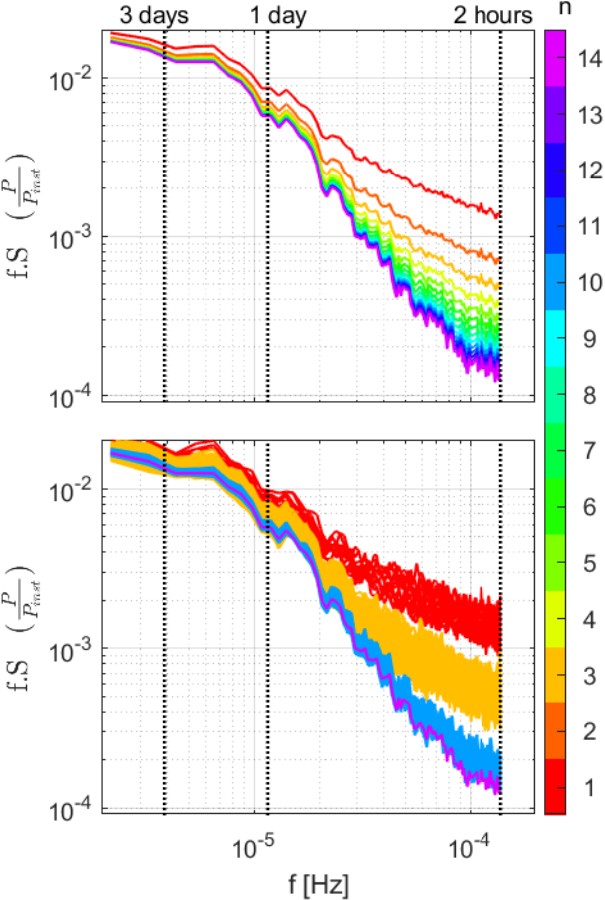

**Figure 7.** PSDs for up to 14 lumped hourly wind power output time series per installed capacity ($\frac{P}{P_{inst}}$). All computations consider an equal distribution of the installed capacity over the considered sites. Colors indicate the number of lumped time series (n). Bottom panel: all possible combinations for n = [1,3,10,14]. Top panel: averaged PSD of all possible combinations for n = [1,2,...,14].

## 4.4 Optimization of wind farm positioning

As observed from Fig. 7, the same number of lumped wind farm time series can display different PSD, depending on which individual time series are aggregated.

By extracting wind farm combinations with the smallest integrals of the PSD with respect to frequencies between $(3\ \mathrm{h})^{-1}$ and $(2\ \mathrm{h})^{-1}$, the wind farm combinations with the lowest fluctuations are obtained. Figures 8 and 9 give examples of optimized wind farm combinations for $n = 4$ and $n = 7$, respectively. For reference, the worst wind farm combinations are also given, those with the highest integrals. It is observed that the wind farms appearing in the optimized combinations, i.e. with the smallest integrals, are scattered, while the opposite is observed for the worst combinations.

The 5th and 95th percentiles and the standard deviation of the hourly step-change function of the lumped time series for each example are also given in Fig. 8 and Fig. 9. Both the percentiles and the standard deviations of the optimized combinations are considerably lower compared with the worst combinations, around 15% and 18%, respectively, for $n = 4$ and around 13% and 15%, respectively, for $n = 7$. In addition, the percentiles and standard deviation of the hourly step-change function of the lumped time series for all combinations of $n < 14$ wind farms are given in Table 3 together with the corresponding percentiles and standard deviation of the lumped time series with the smallest integral.

It can be concluded that the combinations of individual site locations when building $n$ wind farms in the Faroe Islands has a considerable impact on the hourly fluctuations of the total power output time series, information that ought to be of interest to operators. Wind farm portfolios with distant sites are preferred in order to have the most stable lumped wind power time series.

## 4.5 Coherency between pairs of wind farm power output data sets at various frequency ranges

The focus of this wind farm optimization study has been to minimize the two-three hourly fluctuations of the overall power output time series. As argued in the Introduction, and observed in the results, optimized wind farm portfolios yield lower fluctuations at these frequencies. For longer periods, longer than a few days, the results in Sect. 4.3 established that there is a limited smoothing effect for any wind farm combinations in the region. The focus of this subsection is on the smoothing effect for frequencies in between these two.

High wind speeds are associated with high standard deviations of the time series, thus high spectral amplitudes, as the integral of the spectral amplitudes over frequency equals about half of the variance of the time series. As the spectral amplitudes are higher for the lower frequencies in the considered frequency range, it can be expected that the PSD with the smallest spectral amplitudes at the lower frequencies are correlated with time series with smaller standard deviations and smaller wind speeds, thus smaller capacity factors. This is not desirable, as minimizing the fluctuations of the overall wind power output time series should not be at the cost of lower power production. Therefore, instead of looking at the PSD, this subsection will focus on the coherence function between pairs of wind turbine power output time series.

There is a connection between the coherency between time series and the smoothing effect of their combined time series. The higher the coherence, the lower the smoothing. For a coherence value of one, there is no smoothing of the combined time

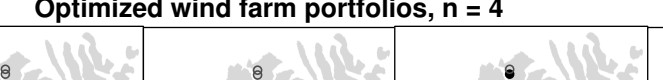

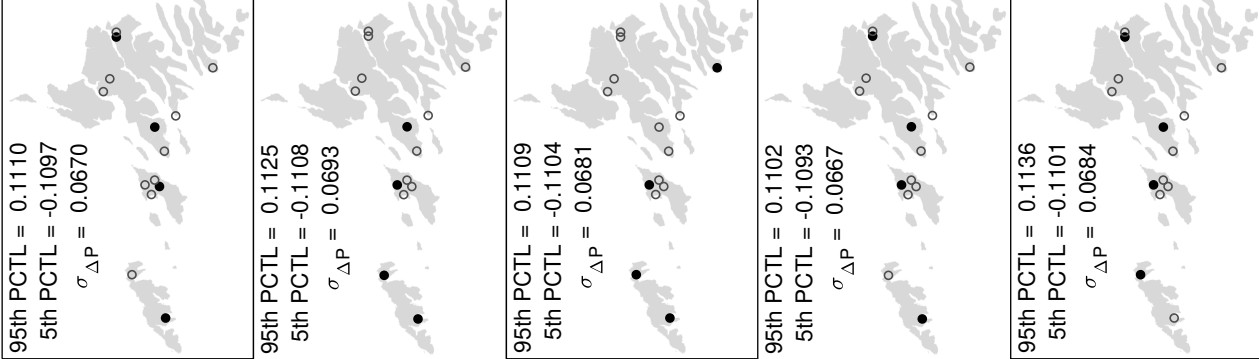

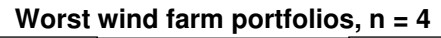

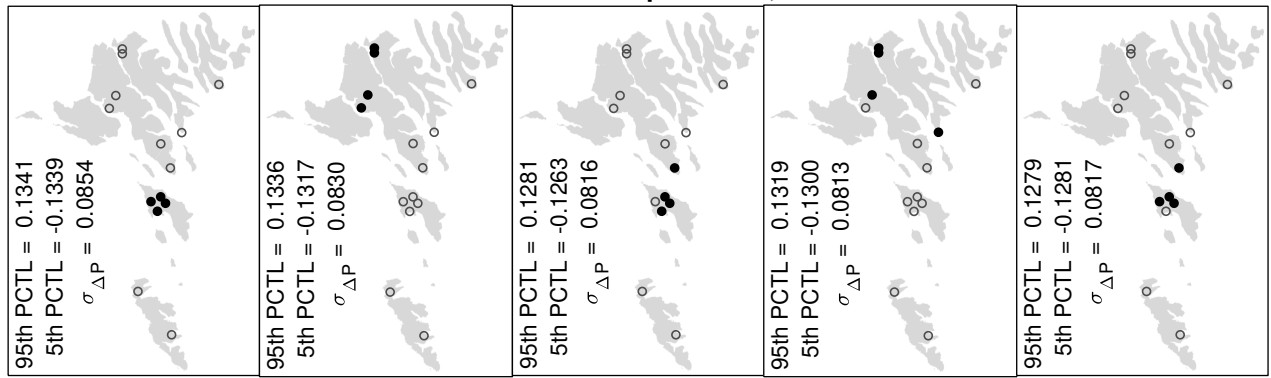

**Figure 8.** Examples of the optimized (top) and the worst (bottom) wind farm portfolios with four wind farms (black filled circles) where the rated power at each wind farm is the same. The open circles mark the locations that are not selected. Theses examples are the portfolios with smallest (optimized portfolios) and largest (worst portfolios) integrals of the PSD with respect to frequencies from $(3\ \text{h})^{-1}$ to $(2\ \text{h})^{-1}$ out of all possible combinations of four wind farms. The 5th and 95th percentiles (PCTL) and the standard deviation of the hourly step-change function of the total wind power time series per installed power is given in the bottom left corner of each example.

series, while a coherence of zero means that the time series are uncorrelated, and the fluctuations of the combined time series will be lower with respect to the produced power.

Figure 10 displays the squared coherence functions of the turbine power output time series for all possible site-pairs out of 330 the fourteen favorable wind farm locations marked in Fig. 2 with respect to inter-site distances. At the lower frequencies, the coherences are high, consistent with the fact that the same weather systems travel across the entire region. As the frequencies increase, the coherence values decrease, which is why from Fig. 7 in Sect. 4.3, it was observed that the smoothing effect from combining wind farms is more evident for higher frequencies. Another general characteristic that can be observed from Fig. 10 is that the closer the inter-site distances are, the higher their coherence values are, which is consistent with the fact that

**Optimized wind farm portfolios, n = 7**

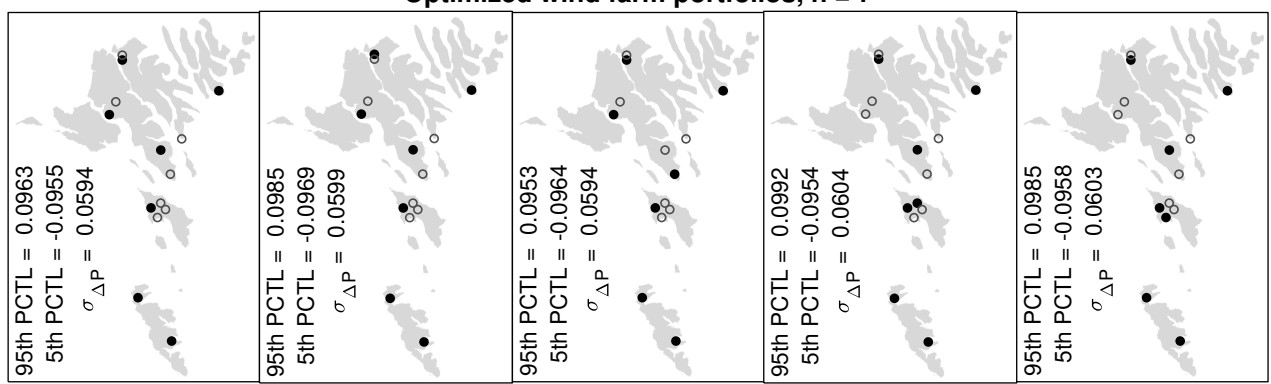

**Worst wind farm portfolios, n = 7**

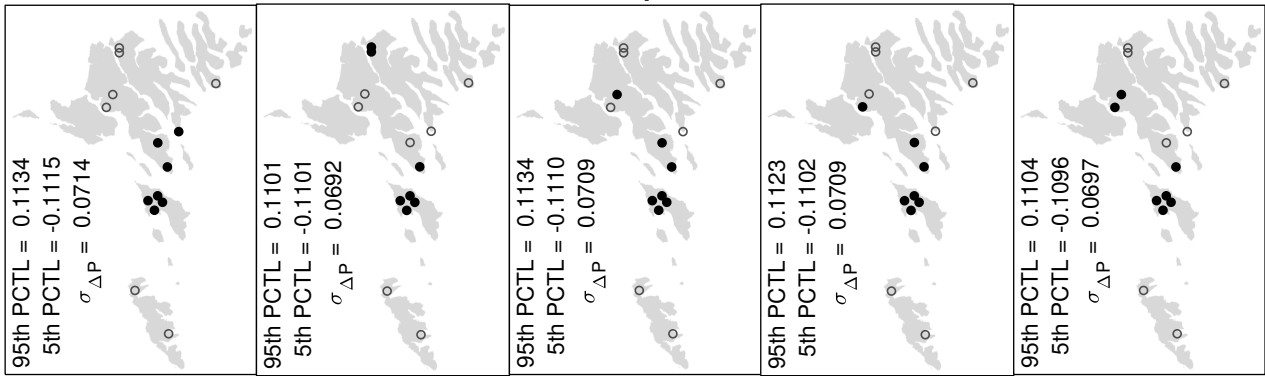

**Figure 9.** Examples of the optimized (top) and the worst (bottom) wind farm portfolios with seven wind farms (black filled circles) where the rated power at each wind farm is the same. The open circles mark the locations that are not selected. Theses examples are the portfolios with smallest (optimized portfolios) and largest (worst portfolios) integrals of the PSD with respect to frequencies from $(3\,\mathrm{h})^{-1}$ to $(2\,\mathrm{h})^{-1}$ out of all possible combinations of seven wind farms. The 5th and 95th percentiles (PCTL) and the standard deviation of the hourly step-change function of the total wind power time series per installed power is given in the bottom left corner of each example.

335    the wind flow has had less time to change when traveling between two closer wind farm sites, which is why in Sect. 4.4, the optimized wind farm portfolio is a combination of distant wind farms.

    To examine the potential smoothing effect on fluctuations at selective frequencies when combining wind farm power output time series, the average squared coherences between the selected frequency ranges are extracted from Fig. 10 and displayed in Fig. 11 with respect to the inter-site distances between site-pairs. Reasonably, the same general characteristics can be observed:

340    higher coherence values for closer wind farms, higher coherence values at lower frequencies, and decreasing coherence values with increasing frequencies. The average squared coherence values for $(3\,\mathrm{h})^{-1} < f < (2\,\mathrm{h})^{-1}$ decrease rapidly with inter-site distance and are already below 0.02 for all pairs of time series with inter-site distances larger than 10 km. This means that in order to minimize the 2-3 hourly fluctuations of aggregated wind farm power output time series, all wind farms in the portfolio

**Table 3.** Range of the 5th (second column) and 95th (third column) percentiles (PCTL) and the standard deviation (fourth column) of the hourly step-change function of the lumped power output time series for all possible combinations of $n$ wind farms (first column) per installed wind power capacity; the wind farm capacities are equally distributed over all considered sites. The 5th (fifth column) and 95th (sixth column) percentiles and the standard deviation (seventh column) of the hourly step-change function for the combinations with the smallest spectral integral with respect to frequencies from $(3\,\mathrm{h})^{-1}$ to $(2\,\mathrm{h})^{-1}$.

| n | 5th PCTL | 95th PCTL | $\sigma$ | 5th PCTL$_{\mathrm{OPT}}$ | 95th PCTL$_{\mathrm{OPT}}$ | $\sigma_{\mathrm{OPT}}$ |
|---|---|---|---|---|---|---|
| 1 | [-0.2013;-0.1659] | [0.1668;0.2089] | [0.1075;0.1277] | -0.1682 | 0.1671 | 0.1080 |
| 2 | [-0.1793;-0.1301] | [0.1288;0.1789] | [0.0817;0.1096] | -0.1339 | 0.1354 | 0.0863 |
| 3 | [-0.1475;-0.1174] | [0.1173;0.1488] | [0.0717;0.0920] | -0.1198 | 0.1255 | 0.0761 |
| 4 | [-0.1339;-0.1086] | [0.1091;0.1341] | [0.0660;0.0854] | -0.1097 | 0.1110 | 0.0670 |
| 5 | [-0.1251;-0.1017] | [0.1029;0.1248] | [0.0630;0.0797] | -0.1043 | 0.1052 | 0.0649 |
| 6 | [-0.1173;-0.0974] | [0.0986;0.1201] | [0.0606;0.0756] | -0.0975 | 0.0999 | 0.0607 |
| 7 | [-0.1118;-0.0941] | [0.0953;0.1134] | [0.0592;0.0714] | -0.0955 | 0.0963 | 0.0594 |
| 8 | [-0.1068;-0.0928] | [0.0939;0.1094] | [0.0581;0.0679] | -0.0950 | 0.0960 | 0.0591 |
| 9 | [-0.1030;-0.0917] | [0.0926;0.1050] | [0.0575;0.0655] | -0.0919 | 0.0938 | 0.0578 |
| 10 | [-0.1011;-0.0906] | [0.0919;0.1022] | [0.0571;0.0634] | -0.0924 | 0.0936 | 0.0571 |
| 11 | [-0.0985;-0.0902] | [0.0913;0.0995] | [0.0568;0.0619] | -0.0907 | 0.0923 | 0.0568 |
| 12 | [-0.0961;-0.0905] | [0.0908;0.0971] | [0.0566;0.0603] | -0.0911 | 0.0908 | 0.0570 |
| 13 | [-0.0937;-0.0900] | [0.0913;0.0940] | [0.0566;0.0587] | -0.0910 | 0.0913 | 0.0566 |

should be placed at least 10 km apart from each other. In order to do so in a small region, and at the same time include several wind farms in the portfolio, most of the spatial area in the region has to be utilized. Although comparably less, the average squared coherence for $(9\,\mathrm{h})^{-1} < f < (6\,\mathrm{h})^{-1}$ also decreases rapidly with inter-site distance, reaching a value of around 0.05 for an inter-site distance of 30 km. This means that there are still possibilities for a maximum smoothing effect on the 6-9 hourly fluctuations for optimized wind farm portfolios consisting of a few wind farms, keeping in mind that the largest inter-site distances are around 90 km.

The average squared coherence values for $(24\,\mathrm{h})^{-1} < f < (16\,\mathrm{h})^{-1}$ and $(72\,\mathrm{h})^{-1} < f < (48\,\mathrm{h})^{-1}$ decrease comparably more slowly with frequency, and the values are scattered. The lowest average squared coherence values for $(24\,\mathrm{h})^{-1} < f < (16\,\mathrm{h})^{-1}$ and $(72\,\mathrm{h})^{-1} < f < (48\,\mathrm{h})^{-1}$ are 0.16 and 0.36, respectively. Although a smoothing effect at these frequencies should be observed for combinations of two time series with low coherence values, combining more than a couple of wind farms into a wind farm portfolio is assumed to result in a limited smoothing effect at these frequency ranges in the given region. Moreover, optimized wind farm portfolios consisting of more than a couple of wind farms are speculated to be determined by the standard deviation of the power output time serious at the expense of the total power production, rather than by a combination of wind farms with low coherence values. However, the trend of a decrease in the average squared coherence values with inter-site

distance is noticeable. By extrapolating, it is expected that optimized wind farm portfolios consisting of multiple wind farms could have a high daily smoothing effect across larger regions without a reduction in the total power production.

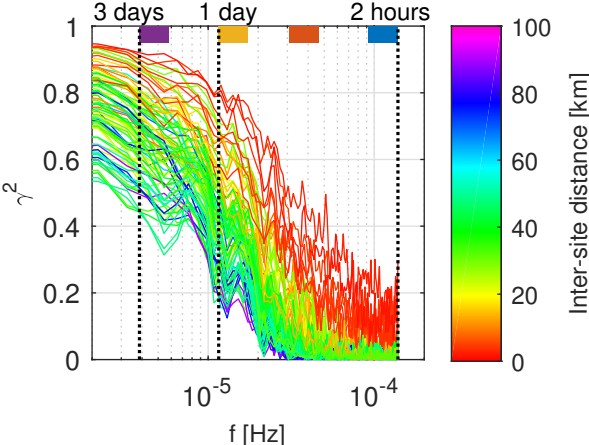

**Figure 10.** Squared coherence functions for all possible site-pairs out of the fourteen turbine power output time series placed at favorable wind farm locations marked in Fig. 2 (total of 91 pairs). The colors of the coherence functions represent the distances in km between site-pairs, see color-bar. The four colors marked at the top of the figure are added to mark the frequency ranges considered in Fig. 11.

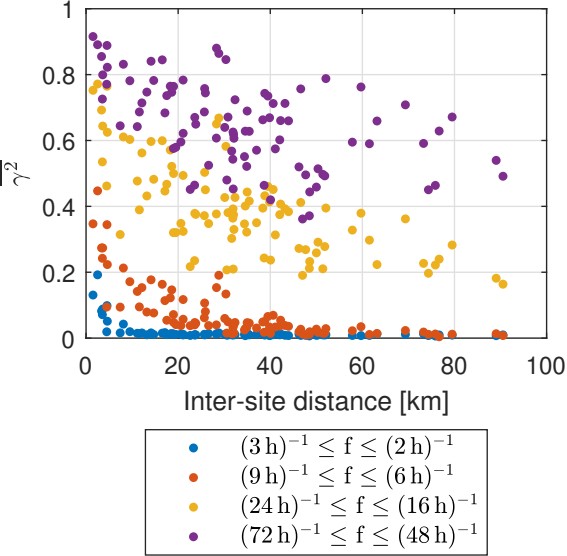

**Figure 11.** Average squared coherence for frequencies between $(3\,h)^{-1}$ and $(2\,h)^{-1}$ (blue color), $(9\,h)^{-1}$ and $(6\,h)^{-1}$ (red color), $(24\,h)^{-1}$ and $(16\,h)^{-1}$ (yellow color), and $(72\,h)^{-1}$ and $(48\,h)^{-1}$ (purple color) with respect to inter-site distance for all site-pairs considered in Fig. 10. The considered frequency ranges are marked in Fig. 10.

# 5   Conclusions

The identification of the best distribution of wind farm production within a region has mainly been done in terms of identifying the energy yield. However, the present study discusses the use of knowledge on the underlying spatial-temporal characteristics of the governing wind field to assist in the search on system configurations for assuring a reduced variability in the power generation. The natural smoothing effect from spatially distributed wind farm sites in a small island system with a complex terrain is investigated in this study, using the Faroe Islands as a case study. For this, hourly modeled wind turbine power output data 45 m a.g.l. are analyzed; converted from WRF-generated wind speed data at fourteen favorable wind farm site locations as selected by Magnussen (2017). All results are presented per installed wind power capacity, i.e., normalized with respect to the rated power.

PSD, hourly step-change functions, and duration curves are generated, and the 5th and 95th percentiles and the standard deviations of the hourly step-change functions are extracted. As expected based on the literature, the smoothing from lumped power output time series is evident, with smaller high frequency PSD amplitudes, lower hourly fluctuations, and fewer periods with zero and rated power production compared with single wind turbine outputs.

By optimally distributing wind farms, spatial smoothing can be maximized, resulting in less regulation needed from the operator. The focus of this study is on the smoothing effect on the 1-3 hourly scale. With the composition of the local power grid, it is considered that limiting the inherent wind-variability on the hourly timescale is of local interest, especially as installed wind power is expected to increase rapidly in the future. In addition, it is argued that the hourly coherency between wind farm power outputs in a small region is expected to be dependent on how the regional weather travels between local sites, while e.g., pairs of wind farm power output data are expected to be uncorrelated at the scale of seconds or minutes, and close to fully coherent at the scale of days due to the same weather regime being present across the country. This study presents optimized wind farm portfolios for the Faroe Islands, where the objective function is set to be the integral of the PSD for frequencies between $(3\text{ h})^{-1}$ and $(2\text{ h})^{-1}$, thus minimizing 2-3 hourly fluctuations.

Wind farm capacities at fourteen pre-defined wind farm site locations are optimized. The results show that the wind farm capacities at remote sites should be higher, and that the wind farm capacities at clustered sites should be lower. However, the optimization has only a small influence on the hourly fluctuations compared with if the wind farm capacities at the pre-defined wind farm site locations were equally distributed over all wind farms. The decrease in the 5th and 95th percentiles and standard deviation of the hourly step-change function is about 2%.

The optimization algorithm is performed for two additional cases. Case II: excluding two distant site locations - located ≥25 km apart from the rest of the sites. Case III: setting upper and lower boundaries for the installed wind power capacity at each site. Also here, the optimization has only a small influence on the hourly fluctuations compared with if the wind farm capacities at the pre-defined wind farm site locations were equally distributed over all wind farms. However, comparing case II to the first case, the 5th and 95th percentiles and standard deviation of the hourly step-change function are increased by 7%. The increase is found to be more pronounced when the two distant sites are excluded compared with if any other two site locations were to be excluded instead, demonstrating the importance of the smoothing effect from distant sites.

As the Faroe Islands consist of a limited spatial area surrounded by ocean far from any other land, the possibilities for the spatial distribution of wind farms are limited. To examine the achievable smoothing effect, PSDs are generated for up to 14 aggregated wind turbine time series. Results show less smoothing as more wind farms are integrated. The smoothing effect is most evident at the highest frequencies, but noticeable for periods of up to 1-2 days. It is also seen that the high frequency fluctuations are highly dependent on which of the individual site locations are considered. Optimized wind farm combinations, in order to minimize the 2-3 hourly power output fluctuations, are wind farms that are located distant from each other, while the worst wind farm combinations consist of clustered wind farms. When building wind farms in four out of the fourteen favorable wind farm areas, optimized wind farm positioning decreases the 5th and 95th percentiles and the standard deviation of the hourly step-change function by 15% and 18%, respectively, compared with the worst wind farm combinations.

By examining the coherency of the power output data between pairs of wind farms, it is concluded that a general characteristic in order to reach a maximum smoothing effect on the hourly scale is to place wind farms $\geq 10$ km apart from each other. Although the focus of this study is primarily on the hourly scale, the coherency between pairs of wind farms focusing on a variety of lower frequency ranges is also presented and used to discuss the potential optimal smoothing effect for the respective frequencies. As hypothesized, coherency on the scale of days is high for all pairs of wind farms, implying that it is inefficient to conduct the optimization algorithm with a focus on minimizing daily scale fluctuations. However, a decreasing trend in the daily coherency is observed with respect to distance between site-pairs, indicating that an optimization algorithm minimizing daily scale fluctuations could be applied for larger regions.

The importance of choosing the best wind farm sites is emphasized in this study in order to naturally balance wind power fluctuations. This feature should be of interest to an operator, as smoother time series result a lower operating effort for the power grid.

This study exclusively considers wind power data 45 m a.g.l., being the hub height of all operating turbines in the Faroe Islands at the time of the preparation of this manuscript. However, future planned wind farms in the region consist of taller wind farms. In fact, six turbines with hub heights of 92 m a.g.l. have recently been installed and started producing electricity for the power grid. The presented work cannot be generalized without further calculations to wind power production at this or other heights, as wind is a function of height, and so are the time series fluctuations and their spatial coherencies. A suggestion for a future study is to include diverse wind turbine models in the analysis in order to investigate if and how these affect the results.

**Appendix A: Filtering for aliasing effect**

The WRF-generated wind speed dataset used in this study display artificial high frequency fluctuations from the aliasing effect (Poulsen et al., 2021). Thus, before usage, these time series must be corrected: For each WRF-generated wind speed time series, the power spectral density is generated and corrected for the aliasing effect using the method given in Kirchner (2005), with the parameters $fc = 1$ and $f\_limit = \frac{1}{1 \text{ day}}$, see Fig. A1. By reversing the corrected spectral calculations and preserving the phase of the discrete Fourier transform of the original time series, a new de-aliased time series is generated. Finally, the

average wind speed value of the original time series is added to the de-aliased time series. Figure A2 displays an example of a scatter-plot between the WRF-generated wind speed time series before and after de-aliasing of the time series.

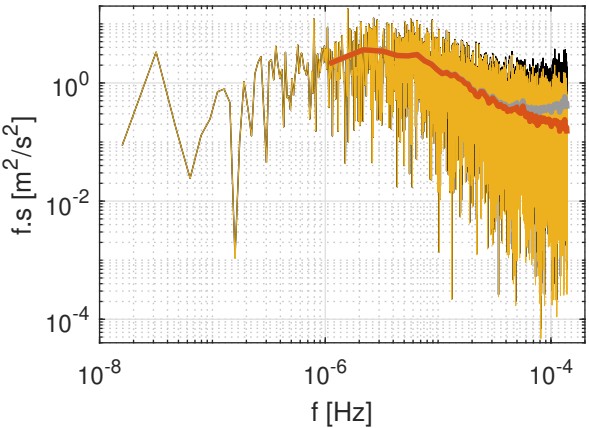

**Figure A1.** Black line represents raw PSD calculated from the WRF-generated wind speed time series at site C. Yellow superimposed line is the corresponding PSD corrected for the aliasing effect using the method given in Kirchner (2005). Note that the black line is behind the yellow line. The gray and red lines represent the averaged PSDs from the black and yellow lines, respectively.

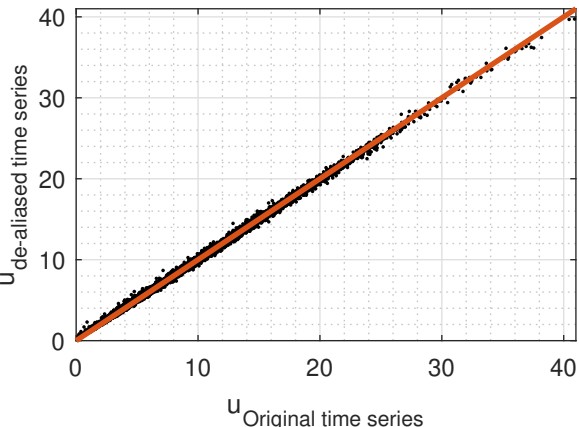

**Figure A2.** Scatter plot between WRF-generated wind speed time series at site C before (x-axis) and after (y-axis) de-aliasing of the time series.

*Author contributions.* TP generated the results and wrote the initial manuscript. HGB supervised. All authors provided important input through discussion, feedback, and revision of the manuscript.

*Competing interests.* The authors declare that they have no conflicts of interest.

*Acknowledgements.* The authors are grateful to the local power company SEV for giving access to measurements and WRF-generated data. We thank Kjeller Vindteknikk for their fast response to our questions regarding the WRF-generated data. TP acknowledges Equinor for their financial support of her Ph.D project "Analysis and modelling of the wind energy resources in the Faroe Islands".

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
