# Peer review of "Optimization of wind farm portfolio to minimize the for minimizing overall power fluctuations at selective frequencies - a case study for of the Faroe Islands"

_Wind Energy Science, 2022_

## Author Comment (AC1)

**Overall quality:**

Interesting work, the case study is relevant and the proposed procedure has been well described. The text is quite clear and easy to follow.

**Thank you very much for taking the time to evaluate our manuscript. Your comments are much appreciated.**

In this file, the black colored text is your comments. We have done our best to response/answer all your comments, which is given as green colored text.

Because both reviewers are asking to get more explanation on why we have focused on the 2-3 h fluctuations and not other frequency ranges, this has been elaborated on in the text. In addition, one additional subsection is included, discussing potential smoothing effect at other frequency ranges (subsection 4.5, see end of this document).

However, I see two important points to improve:

> The authors should elaborate on why the frequency interval from $1/2\ h^{-1}$ to $1/3\ h^{-1}$ has been selected for the optimization instead of larger periods (e.g. 1-2 days) or a wider range of frequencies (e.g. from daily to $1/2\ h^{-1}$). This choice should be carefully motivated since it is at the article's core. One could argue that the 2-3h period falls right within the spectral gap of typical wind variations at a site and therefore relatively low variability is expected in that range. Valid arguments justifying the frequency range selection should be presented already in the introduction (e.g. around lines 31 or 59) and then renewed in the discussion sections (e.g. around lines 157 and 237).
>
> Your comment makes much sense.  Arguments are added around line 31. And again, shortly mentioned around lines 157 (Method>Optimization Method) and 237 (discussion).

- The authors should present at least a hypothesis for why the optimization provides a limited benefit compared to an even distribution of the capacity. Moreover, it would be nice to see some comments about what would be expected if the same methodology was applied to different case studies.

To answer your comment, the following discussion is added to section 4.2. (The end of section 4.2 has been removed to a separate subsection (4.2.1)):

> The reduction of the 5th and 95th percentiles and standard deviations of the hourly step-change functions are 2% or less for the three optimized cases. It may be speculated that the observed limited hourly smoothing effect is because the improvement that the optimization adds is small compared to the smoothing that already occurred by considering the equally weighted spatially distribution of the wind farms.
>
> To investigate this further, small wind farm portfolios in the region are examined, by looking at combinations of three wind farms. With the combination of the southernmost site and the two clustered northernmost sites, optimized wind farm capacity distribution places around 27% of the total power capacity of the wind farm portfolio at each of the northernmost sites and almost 47% at the southernmost site. The distribution is logical, with lower capacities at both of the clustered wind farm sites, and higher capacity at the distant site. If the two clustered sites were fully correlated on 2-3 hourly scale while the distant site was uncorrelated, the capacity distribution is expected to be 25%, 25%, and 50%, respectively, which is close to what is observed for this set-up. For the optimized portfolio with this set-up, the 5th and 95th percentiles and the standard deviation of the hourly power output step-change function are 4-5% lower compared to if the wind power capacity were to be equally distributed over the three sites. Comparing the optimized wind farm portfolio to a portfolio where 40% of the total wind power capacity are placed at each of the two northernmost wind farms and 20% at the southernmost wind farm – a rather illogical distribution in terms of spatially smoothing of the power capacity – the optimized wind farm portfolio contains 12-13% lower 5th and 95th percentiles and standard deviation. Indicating that the power capacity distribution of the wind farms matters.
>
> Going back to case I (all fourteen favorable wind farm sites in the region), although the optimized portfolio has only limited effect on the smoothing effect compared to the portfolio where the power capacities are equally distributed, comparing the overall power outputs of the optimized distribution of case I to the power output of the equally distributed capacity of case III with the constrain of the given boundary conditions, the 5th and 95th percentiles and the standard deviation of the hourly step-change function are 4% lower. Likewise, by comparing the power output of the optimized distribution of case I to a set-up where 11% of the total wind power capacity is placed at the four closest clustered sites as well as at the two northernmost sites, and 4.5% at the rest of the sites (with the intent to distribute the capacity of the wind farms poorly), the 5th and 95th percentiles and the standard deviation of the hourly step-change function are 6% and 8% lower, respectively. Indicating that the wind power capacity distribution matters.
>
> It is expected that the results would be different for other case studies. The magnitude of the smoothing due to spatial distribution of wind farm capacities depends on the magnitude of the coherences and the details of the shape of the PSD curves. Other locations with e.g. simpler terrain are expected to have larger coherencies. On the other hand, larger regions have the advantage to be able to scatter wind farms to a larger extend.

**Specific comments:**

- Title: "… to minimize the overall power fluctuations …" – maybe it would be better to rephrase it a bit, e.g. with something like: "… minimize the power fluctuation at selected frequencies …" This more accurate. Thank you for your suggestion. The title is being changed to: "Optimization of wind farm portfolio to minimize the overall power fluctuations at selected frequencies - a case study for the Faroe Islands"

- Abstract: "The focus is mainly on the smoothing effect in highest resolvable frequencies." – it would be nice to add a brief explanation for why these frequencies are relevant.

  The following is added to the sentence: *"The focus is mainly on the smoothing effect in the 1-3 hourly scale, where the coherency between wind farm power outputs is expected to be dependent on how the regional weather travels between local sites, making optimizations of wind farm portfolios relevant – in oppose to a focus on either lower or*

*higher frequencies in the scale of days or minutes, respectively, where wind farm power output time series are expected to be either close to fully coherent due to the same weather conditions covering a small region or not coherent as the turbulence for separate wind farm locations are expected to be uncorrelated. Results show that an optimization of …"*

- Abstract: "decrease the 1-3 hourly fluctuations considerably" – it would be better to be more quantitative.

  Understood. That was actually the intent of the last line in the abstract. For clarification, more information is added. The end of the abstract is changed to the following:

  *"However, choosing optimized combinations of the individual wind farm site locations decreases the 1-3 hourly fluctuations considerably. For example, selecting a portfolio with four wind farms (out of the fourteen pre-defined wind farm site locations) results in 15% lower 5th and 95th percentiles of the hourly step change function when choosing optimal wind farm combinations compared to choosing the worst wind farm combinations. For an optimized wind farm portfolio of seven wind farms, this number is 13%. Optimized wind farm portfolios consist of distant wind farms, while the worst portfolios consist of clustered wind farms."*

- Line 94 – The authors should explain why that height was selected. Is it the turbine's hub height? Yes. This height is chosen, because all currently operating wind turbines in the Faroe Islands have a hub height of 45 m a.g.l. (this explanation is added in the revised version). (Additional information: currently constructed wind farms/future planned wind farms consist of/will consist of taller wind turbines. This information is not considered in this study, but instead, given as future study suggestions at the end of the manuscript).

- Line 96 – Please motivate the choice of this turbine and give at least some minimum specifics like the hub height and the rated power.

  OK. Additional information is added to this paragraph (but the hub height is added to the paragraph above):

  *"The wind speed time series are modeled to power output time series using the power curve of an Enercon E-44 wind turbine with storm control function (Enercon, 2012), having a rated power of 0.9 MW. This turbine model is chosen because most of the currently operating wind turbines in the Faroe Islands are of the type Enercon E-44 (25 out of 28 turbines)."*

- Line 108 – Please indicate the overall duration of the signal and the chunks.
  The following paragraph is added:

  *"The length of the chunks is a compromise between the accuracy of the PSD estimates (smaller chunks, i.e., more chunks) and the frequency resolution and the lowest resolvable frequency (longer chunks). In this study, a length of 256 data points is chosen (10 days and 16 hours), giving a number of 135 overlapping chunks for two-year hourly time series. The PSD estimates will therefore be generated for frequencies between $(256\,h)^{-1}$ (Thus, including PSD estimates for the 3-4 day period of the time scale of migratory low-pressure systems at mid and high latitudes) and the Nyquist frequency of $(2\,h)^{-1}$, with a resolution of $(256\,h)^{-1}$."*

- Line 210 – It would be nice to link this to the findings of previous studies. This sentence is added: *"These characteristics are similar to results observed by e.g. Katzenstein et al. (2010) and Beyer et al. (1993), who analyzed the variability of interconnected wind power time series, spatially dispersed in the area of Texas and northwest Germany, respectively."*
- Line 269 – The paragraph on future work is a bit superficial. Possibly it should be extended or at least better argumented. Agreed. Initially, we suggested two future works. Now the outlook focuses only on the first suggestion, with more details and argument for why this given future work is interesting.

**Technical corrections:**

- The labels or at least the captions of the PSD plots should specify what quantity is considered and its physical dimensions (or if normalized it should be mentioned). Also, I think that adding gridlines to the plots would help their interpretability.

  Grid lines are added to all figures (excluding the maps). All time series are normalized with respect to the installed wind power capacity (except in the appendix). All the captions for these figures now mention that they represent "*hourly wind power output time series per installed capacity* $\left(\frac{P}{P_{inst}}\right)$". And $\left(\frac{P}{P_{inst}}\right)$ is added to the label of all PSD plots.

- I could find several typos and a few small mistakes in the use of English. I will only list a few here, but please check the manuscript thoroughly before submitting the revised version. Thank you. The below typos have been corrected, and so are additional typos and mistakes found when going through the manuscript again.
- Abstract: "5th and 95th percentiles" – specify "of the hourly step-change functions". Thanks! "of the hourly step-change functions" is now specified in the abstract.
- Line 37 – extent Thanks, this is now corrected
- Line 42 – recent Thanks, this is now corrected
- Line 58 – geography Thanks, this is now corrected
- Line 60 – is available I cannot find this in the manuscript
- Line 162 – constraint Thanks, this is now corrected
- Line 210 – pronounced Thanks, this is now corrected
- Line 270 - planned Thanks, this is now corrected

**4.5 A note on the coherency between pairs of wind farm power output data at various frequency ranges**

290 The focus of this wind farm optimization study has been to minimize the two-three hourly fluctuations of the overall power output time series. As argued in the Introduction, and observed in the results, optimized wind farm portfolios yield less fluctuations at these frequencies. For longer periods, longer that a few days, results in section 4.3 established that there were limited smoothing effect for any wind farm combinations in the region. The focus of this subsection is on the smoothing effect for frequencies in between these two.

295 High wind speeds are associated with high standard deviations, thus high spectral values, as the integral of the spectral values over frequency equals about half of the variance of the time series. And since the spectral values are higher for lower the frequencies, in the considered frequency range, it can be expected that the PSD with the smallest spectral values at the lower frequencies are correlated to time series with smaller standard deviations and smaller wind speeds, thus smaller capacity factors. This is not desirable, as minimizing the fluctuations of the overall wind power output time series should not be at the 300 cost of lower power production. Therefore, instead of looking at the PSD, this subsection will focus on the coherence function between pairs of wind turbine power output time series.

There is a connection between the coherency between time series and the smoothing effect of their combined time series. The higher the coherence, the less the smoothing. For a coherence value of one, there will be no smoothing of the combined time series, while a coherence of zero means that the time series are uncorrelated, and the fluctuations of the combined time 305 series will be less with respect to produced power.

Fig. 10 displays the squared coherence functions for all possible site-pairs of turbine power output time series, out of the fourteen favourable wind farm locations locations marked in Fig. 2, with respect to inter-site distances. At the lower frequencies, the coherences are high, consistent with that the same weather systems travel across the entire region. As the frequencies increase, the coherence values decrease, which is why from figure 7 in subsection 4.3, it was observed that the 310 smoothing effect from combining wind farms is more evident for the higher frequencies. Another general characteristics that can be observed from Fig. 10 is that the closer the inter-site distances are, the higher their coherence value are, consistent with that the wind flow has had less time to change when traveling between two closer wind farm sites, which is why in subsection 4.4, optimized wind farm portfolio are combinations of distant wind farms.

To examine the potential smoothing effect for fluctuations at selective periods when combining wind farm power output time 315 series, the average squared coherences between selected frequency ranges are extracted from Fig. 10 and displayed in Fig. 11 with respect to inter-site distances between site-pairs. As expected, the same general characteristics can be observed: higher coherence values for closer wind farms, higher coherence values at lower frequencies, and decreasing coherence values with increasing frequencies. The average squared coherence values for $\frac{1}{3h} < f < \frac{1}{2h}$ decrease rapidly with inter-site distance and are already below 0.02 for all pair of time series having inter-site distances larger than 10 km. This means that in order to minimize 320 the 2-3 hourly fluctuations of aggregated wind farm power output time series, all wind farms in the portfolio should be placed at least 10 km apart from each other. In order to do so in a small region, and at the same time include several wind farms in the

portfolio, most of the spatial area in the region has to be utilized. Although comparable less, the average squared coherence for $\frac{1}{9h} < f < \frac{1}{6h}$ also decrease rapidly with inter-site distances, reaching a value around 0.05 for an inter-site distance of 30 km. This means that there are still possibilities for a maximum smoothing effect of 6-9 h hourly fluctuations for optimized wind

325  farm portfolios consisting of a few wind farms, having in mind that the largest inter-site distances are around 90 km.

The squared coherence values for $\frac{24}{3h} < f < \frac{1}{16h}$ and $\frac{1}{72h} < f < \frac{1}{48h}$ decrease comparably more slowly with frequency, and the values are scattered. The lowest squared coherence values for $\frac{24}{3h} < f < \frac{1}{16h}$ and $\frac{1}{72h} < f < \frac{1}{48h}$ are 0.16 and 0.36, respectively. Although, a smoothing effect at these frequencies should be observed for combinations of two time series with low coherence values, combining more that a couple of wind farms in a wind farm portfolio is assumed to have limited

330  smoothing effect for the these frequency ranges in the given region. And optimized wind farm portfolios consisting of more than a couple of wind farms are speculated to be determined by the standard deviation of the power output time serious at the expense of the total power production, rather than the combination of wind farms with low coherence values. The trend of the decrease of the squared coherence values with inter-site distance is is observable. By extrapolating, it is expected that optimized wind farm portfolios consisting of multiple wind farms could have a high daily smoothing effect at larger regions

335  without the expense of total power production.

[Figure]

**Figure 10.** Squared coherence functions for all possible site-pairs out of the fourteen turbine power output time series, placed at the favourable wind farm locations marked in Fig. 2 (a total of 91 pairs). The colors of the coherence functions represent the distances in km between site-pairs, see colorbar. The four colors marked at the top of the figure are added to mark the frequency ranges considered in Fig. 11.

[Figure]

**Figure 11.** The average squared coherence for frequencies between $(3\ h)^{-1}$ and $(2\ h)^{-1}$ (blue color), $(9\ h)^{-1}$ and $(6\ h)^{-1}$ (red color), $(24\ h)^{-1}$ and $(16\ h)^{-1}$ (yellow color), and $(72\ h)^{-1}$ and $(48\ h)^{-1}$ (purple color), with respect to inter-site distance for all site-pairs considered in figure 10. The considered frequency ranges are marked in figure 10.

20

---

## Author Comment (AC2)

This paper is addressing the minimization of the overall power fluctuations for different wind farm portfolios. In general, the proposed method has the potential for publication in WES. I have the following comments:

**Thank you very much for taking the time to evaluate our manuscript. Your comments are much appreciated.**

In this file, the black colored text is your comments to the manuscript. We have done our best to response/answer all your comments, which is given as green colored text.

Because both reviewers are asking to get more explanation on why we have focused on the 2-3 h fluctuations and not other frequency ranges, this has been elaborated on in the text. In addition, one additional subsection is included, discussing potential smoothing effect at other frequency ranges (subsection 4.5, see the end of this document).

- The 5th and 95th percentiles of the step change function of wind power time series are used as statistical variability metrics in this paper, which from my point of view represent the range of power fluctuations. It could be more descriptive if the authors were also looking at mean and standard deviation.
  Thank you for this comment. The standard deviation is added. However, the mean is excluded, as the mean of the step change function is approximately zero. This is because the power production is always between zero and rated power, thus averaging over hourly increases/decreases for longer periods adds up to approximately zero. Two year of hourly data equals 17520 data points, and the maximum mean value would thus be $\pm 6*10^{-5}$ MW per MW$_{installed}$ (1 MW per MW$_{installed}$ divided by 2*24*365).

- This paper should be restructured to improve its readability.
  The manuscript is now restructured. All but one of your suggestions are applied.
  * The Section Introduction needs to improve. Please follow this sequence: problem definition and motivation for research in this field, literature review, and the main contributions of this research.
  This sequence is applied to the Introduction
  Subsection "A note on ignoring the wind farm smoothing effect" could move to the introduction as the paper assumptions.
  This subsection is moved to the end of the introduction.
  The Section "Data" could merge with the Section "Result".
  Although the suggestion is appreciated, the Sections "Data" and "Result" are still separate, so the reader can more easily locate what data has been used for the study.
  Optimization theory in Subsection "Optimization of wind farm capacities" is better to move to Section "Method".
  Thank you. The optimization theory is moved to the end of the Section "Method"

(in addition, the description of the optimization is also changed according to your comment further down).

- There are language errors and typos, e.g., constrain instead of constraint in pg 8 line 162.

Thank you. We have corrected multiple language errors in the revised version of the manuscript, including the error you mention here.

- Illustrated PSD in Fig. 5 is a little confusing. What are the time step and the time interval for the PSD analysis performed in this figure?

All PSD are generated with the same method. For clarification, the following description is added to the end of section **Method > Spectral Analysis**:

*"The length of the chunks is a compromise between the accuracy of the PSD estimates (smaller chunks, i.e., more chunks) and the frequency resolution and the lowest resolvable frequency (longer chunks). In this study, a length of 256 data points is chosen (10 days and 16 hours), giving a number of 135 overlapping chunks for two-year hourly time series. The PSD estimates will therefore be generated for frequencies between (256 h)$^{-1}$ (Thus, including PSD estimates for the 3-4 day period of the time scale of migratory low-pressure systems at mid and high latitudes) and the Nyquist frequency of (2 h)$^{-1}$, with a resolution of (256 h)$^{-1}$."*

The wording of the caption and label in Fig. 5 are also changed a little bit

- Could the authors bring more details into the mathematical presentation of the optimization objective function represented by eq.2? The PSD of which function is going to be minimized in the specified frequency range.

  Yes. One additional equation is added, describing the power output time series of wind farm portfolios (Eq. (2) in the re-submitted manuscript). It is the PSD of this equation, which is used to derive an optimized portfolio, being the portfolio where the fluctuations of the total wind power output time series are minimized for frequencies between (3 h)$^{-1}$ and (2 h)$^{-1}$. The section describing the optimization has been changed accordingly in the re-submitted manuscript, as an attempt to improve the description.

- Have the authors tested different frequency ranges, and why is the frequency range (2h)-1 (3h)-1 chosen for the optimization?

  The focus is mainly on frequencies between (3 h)$^{-1}$ and (2 h)$^{-1}$. Arguments for why these frequencies are considered are added to the: Abstract, Introduction, Method > Spectral Analysis, and Discussion

We did not test other frequency ranges. However, we have added a subsection, see below, discussing this topic in more details.

**4.5 A note on the coherency between pairs of wind farm power output data at various frequency ranges**

290 The focus of this wind farm optimization study has been to minimize the two-three hourly fluctuations of the overall power output time series. As argued in the Introduction, and observed in the results, optimized wind farm portfolios yield less fluctuations at these frequencies. For longer periods, longer that a few days, results in section 4.3 established that there were limited smoothing effect for any wind farm combinations in the region. The focus of this subsection is on the smoothing effect for frequencies in between these two.

295 High wind speeds are associated with high standard deviations, thus high spectral values, as the integral of the spectral values over frequency equals about half of the variance of the time series. And since the spectral values are higher for lower the frequencies, in the considered frequency range, it can be expected that the PSD with the smallest spectral values at the lower frequencies are correlated to time series with smaller standard deviations and smaller wind speeds, thus smaller capacity factors. This is not desirable, as minimizing the fluctuations of the overall wind power output time series should not be at the

300 cost of lower power production. Therefore, instead of looking at the PSD, this subsection will focus on the coherence function between pairs of wind turbine power output time series.

There is a connection between the coherency between time series and the smoothing effect of their combined time series. The higher the coherence, the less the smoothing. For a coherence value of one, there will be no smoothing of the combined time series, while a coherence of zero means that the time series are uncorrelated, and the fluctuations of the combined time

305 series will be less with respect to produced power.

Fig. 10 displays the squared coherence functions for all possible site-pairs of turbine power output time series, out of the fourteen favourable wind farm locations locations marked in Fig. 2, with respect to inter-site distances. At the lower frequencies, the coherences are high, consistent with that the same weather systems travel across the entire region. As the frequencies increase, the coherence values decrease, which is why from figure 7 in subsection 4.3, it was observed that the

310 smoothing effect from combining wind farms is more evident for the higher frequencies. Another general characteristics that can be observed from Fig. 10 is that the closer the inter-site distances are, the higher their coherence value are, consistent with that the wind flow has had less time to change when traveling between two closer wind farm sites, which is why in subsection 4.4, optimized wind farm portfolio are combinations of distant wind farms.

To examine the potential smoothing effect for fluctuations at selective periods when combining wind farm power output time

315 series, the average squared coherences between selected frequency ranges are extracted from Fig. 10 and displayed in Fig. 11 with respect to inter-site distances between site-pairs. As expected, the same general characteristics can be observed: higher coherence values for closer wind farms, higher coherence values at lower frequencies, and decreasing coherence values with increasing frequencies. The average squared coherence values for $\frac{1}{3h} < f < \frac{1}{2h}$ decrease rapidly with inter-site distance and are already below 0.02 for all pair of time series having inter-site distances larger than 10 km. This means that in order to minimize

320 the 2-3 hourly fluctuations of aggregated wind farm power output time series, all wind farms in the portfolio should be placed at least 10 km apart from each other. In order to do so in a small region, and at the same time include several wind farms in the

19

portfolio, most of the spatial area in the region has to be utilized. Although comparable less, the average squared coherence for $\frac{1}{9h} < f < \frac{1}{6h}$ also decrease rapidly with inter-site distances, reaching a value around 0.05 for an inter-site distance of 30 km. This means that there are still possibilities for a maximum smoothing effect of 6-9 h hourly fluctuations for optimized wind
325  farm portfolios consisting of a few wind farms, having in mind that the largest inter-site distances are around 90 km.

The squared coherence values for $\frac{24}{3h} < f < \frac{1}{16h}$ and $\frac{1}{72h} < f < \frac{1}{48h}$ decrease comparably more slowly with frequency, and the values are scattered. The lowest squared coherence values for $\frac{24}{3h} < f < \frac{1}{16h}$ and $\frac{1}{72h} < f < \frac{1}{48h}$ are 0.16 and 0.36, respectively. Although, a smoothing effect at these frequencies should be observed for combinations of two time series with low coherence values, combining more that a couple of wind farms in a wind farm portfolio is assumed to have limited
330  smoothing effect for the these frequency ranges in the given region. And optimized wind farm portfolios consisting of more than a couple of wind farms are speculated to be determined by the standard deviation of the power output time serious at the expense of the total power production, rather than the combination of wind farms with low coherence values. The trend of the decrease of the squared coherence values with inter-site distance is is observable. By extrapolating, it is expected that optimized wind farm portfolios consisting of multiple wind farms could have a high daily smoothing effect at larger regions
335  without the expense of total power production.

[Figure]

**Figure 10.** Squared coherence functions for all possible site-pairs out of the fourteen turbine power output time series, placed at the favourable wind farm locations marked in Fig. 2 (a total of 91 pairs). The colors of the coherence functions represent the distances in km between site-pairs, see colorbar. The four colors marked at the top of the figure are added to mark the frequency ranges considered in Fig. 11.

[Figure]

**Figure 11.** The average squared coherence for frequencies between $(3\ h)^{-1}$ and $(2\ h)^{-1}$ (blue color), $(9\ h)^{-1}$ and $(6\ h)^{-1}$ (red color), $(24\ h)^{-1}$ and $(16\ h)^{-1}$ (yellow color), and $(72\ h)^{-1}$ and $(48\ h)^{-1}$ (purple color), with respect to inter-site distance for all site-pairs considered in figure 10. The considered frequency ranges are marked in figure 10.

---

## Author Response (AR1)

**Thank you very much for taking the time to evaluate our manuscript. The comments are much appreciated.**

In this file, the black colored text is the reviewers' comments to the manuscript. We have done our best to response/answer all comments, which is given as green colored text.

Because both reviewers are asking to get more explanation on why we have focused on the 2-3 h fluctuations and not other frequency ranges, this has been elaborated on in the text. In addition, one additional subsection is included, discussing potential smoothing effect at other frequency ranges (subsection 4.5 in the revised manuscript).

**Response to RC1**

Interesting work, the case study is relevant and the proposed procedure has been well described. The text is quite clear and easy to follow.

However, I see two important points to improve:

> The authors should elaborate on why the frequency interval from 1/2 $h^{-1}$ to 1/3 $h^{-1}$ has been selected for the optimization instead of larger periods (e.g. 1-2 days) or a wider range of frequencies (e.g. from daily to 1/2 $h^{-1}$). This choice should be carefully motivated since it is at the article's core. One could argue that the 2-3h period falls right within the spectral gap of typical wind variations at a site and therefore relatively low variability is expected in that range. Valid arguments justifying the frequency range selection should be presented already in the introduction (e.g. around lines 31 or 59) and then renewed in the discussion sections (e.g. around lines 157 and 237).
>
> Your comment makes much sense. Arguments are added around line 31. And again, shortly mentioned around lines 157 (Method>Optimization Method) and 237 (discussion).

- The authors should present at least a hypothesis for why the optimization provides a limited benefit compared to an even distribution of the capacity. Moreover, it would be nice to see some comments about what would be expected if the same methodology was applied to different case studies.

  To answer your comment, additional discussion is added to section 4.2. (The end of section 4.2 has been removed to a separate subsection (4.2.1)):

**Specific comments:**

- Title: "… to minimize the overall power fluctuations …" – maybe it would be better to rephrase it a bit, e.g. with something like: "… minimize the power fluctuation at selected frequencies …" This more accurate. Thank you for your suggestion. The title is being changed to: "Optimization of wind farm portfolio to minimize the overall power fluctuations at selected frequencies - a case study for the Faroe Islands"

- Abstract: "The focus is mainly on the smoothing effect in highest resolvable frequencies." – it would be nice to add a brief explanation for why these frequencies are relevant.

The following is added: *"The focus is mainly on the smoothing effect on the 1-3 hourly time scale, during which the coherency between wind farm power outputs is expected to be dependent on how the regional weather travels between local sites, thereby making optimizations of wind farm portfolios relevant; in oppose to a focus on either lower or higher frequencies on the scale of days or minutes, respectively, during which wind farm power output time series are expected to be either close to fully coherent due to the same weather conditions covering a small region or not coherent as the turbulences in separate wind farm locations are expected to be uncorrelated. Results show that ..."*

- Abstract: "decrease the 1-3 hourly fluctuations considerably" – it would be better to be more quantitative.

  Understood. That was actually the intent of the last line in the abstract. For clarification, more information is added. The end of the abstract is changed to the following:

  *"However, choosing optimized combinations of individual wind farm site locations decreases the 1-3 hourly fluctuations considerably. For example, selecting a portfolio with four wind farms (out of the fourteen pre-defined wind farm site locations) results in 15% lower 5th and 95th percentiles of the hourly step-change function when choosing optimal wind farm combinations compared with choosing the worst wind farm combinations. For an optimized wind farm portfolio of seven wind farms, this number is 13%. Optimized wind farm portfolios consist of distant wind farms, while the worst portfolios consist of clustered wind farms."*

- Line 94 – The authors should explain why that height was selected. Is it the turbine's hub height? Yes. This height is chosen, because all operating wind turbines in the Faroe Islands at the time of the preparation of the manuscript had a hub height of 45 m a.g.l. (this explanation is added in the revised version). (Additional information: currently constructed wind farms/future planned wind farms consist of/will consist of taller wind turbines. This information is not considered in this study, but instead, given as future study suggestions at the end of the manuscript).

- Line 96 – Please motivate the choice of this turbine and give at least some minimum specifics like the hub height and the rated power.

  OK. Additional information is added to this paragraph (but the hub height is added in the paragraph above):

  *"The wind speed time series are modeled to power output time series using the power curve of an Enercon E-44 wind turbine with a storm control function (Enercon2012) and a rated power of 0.9 MW. This turbine model is chosen because most of the currently operating wind turbines in the Faroe Islands are of the type Enercon E-44."*

- Line 108 – Please indicate the overall duration of the signal and the chunks. The following paragraph is added:

  *"The length of the chunks is a compromise between the accuracy of the PSD estimates (smaller chunks, i.e., more chunks) and the frequency resolution and the lowest resolvable frequency (longer chunks). In this study, a length of 256 data points was chosen (10 days and 16 hours), giving 135 overlapping chunks for the two-year long hourly time series. The PSD estimates will therefore be generated for frequencies between $(256 \text{ h})^{-1}$ (thus, including PSD estimates for the 3-4 day period of the time scale of migratory low-pressure systems at mid and high latitudes) and the Nyquist frequency of $(2 \text{ h})^{-1}$ with a resolution of $(256 \text{ h})^{-1}$."*

- Line 210 – It would be nice to link this to the findings of previous studies. This sentence is added: *"These characteristics are similar to results observed by e.g. Katzenstein et al. (2010) and Beyer et al. (1993), who analyzed the variability of interconnected wind power time series spatially dispersed in the area of Texas and northwest Germany, respectively."*

- Line 269 – The paragraph on future work is a bit superficial. Possibly it should be extended or at least better argumented. Agreed. Initially, we suggested two future works. Now the outlook focuses only on the first suggestion, with more details and argument for why this given future work is interesting.

**Technical corrections:**

- The labels or at least the captions of the PSD plots should specify what quantity is considered and its physical dimensions (or if normalized it should be mentioned). Also, I think that adding gridlines to the plots would help their interpretability.

  Grid lines are added to all figures (excluding the maps). All time series are normalized with respect to the installed wind power capacity (except in the appendix). All the captions for these figures now mention that they represent *"hourly wind power output time series per installed capacity $\left(\frac{P}{P_{inst}}\right)$"*. And $\left(\frac{P}{P_{inst}}\right)$ is added to the label of all PSD plots.

- I could find several typos and a few small mistakes in the use of English. I will only list a few here, but please check the manuscript thoroughly before submitting the revised version. Thank you. The below typos have been corrected. For the revised manuscript, we have used Wiley Editing Services for proper English language, grammar, punctuation, spelling, and overall style of the manuscript (most of their suggestions).

- Abstract: "5th and 95th percentiles" – specify "of the hourly step-change functions". Thanks! "of the hourly step-change functions" is now specified in the abstract.

- Line 37 – extent Thanks, this is now corrected
- Line 42 – recent Thanks, this is now corrected
- Line 58 – geography Thanks, this is now corrected
- Line 60 – is available I cannot find this in the manuscript
- Line 162 – constraint Thanks, this is now corrected
- Line 210 – pronounced Thanks, this is now corrected
- Line 270 - planned Thanks, this is now corrected

**Response to RC2**

This paper is addressing the minimization of the overall power fluctuations for different wind farm portfolios. In general, the proposed method has the potential for publication in WES. I have the following comments:

- The 5th and 95th percentiles of the step change function of wind power time series are used as statistical variability metrics in this paper, which from my point of view represent the range of power fluctuations. It could be more descriptive if the authors were also looking at mean and standard deviation.
  Thank you for this comment. The standard deviation is added. However, the mean is excluded, as the mean of the step change function is approximately zero. This is because the power production is always between zero and rated power, thus averaging over hourly increases/decreases for longer periods adds up to approximately zero. Two year of hourly data equals 17520 data points, and the maximum mean value would thus be $\pm 6*10^{-5}$ MW per $MW_{installed}$ (1 MW per $MW_{installed}$ divided by 2*24*365).

- This paper should be restructured to improve its readability.
  The manuscript is now restructured. All but one of your suggestions are applied.
  * The Section Introduction needs to improve. Please follow this sequence: problem definition and motivation for research in this field, literature review, and the main contributions of this research.
  This sequence is applied to the Introduction
  Subsection "A note on ignoring the wind farm smoothing effect" could move to the introduction as the paper assumptions.
  This subsection is moved to the end of the introduction.
  The Section "Data" could merge with the Section "Result".
  Although the suggestion is appreciated, the Sections "Data" and "Result" are still separate, so the reader can more easily locate what data has been used for the study.
  Optimization theory in Subsection "Optimization of wind farm capacities" is better to move to Section "Method".
  Thank you. The optimization theory is moved to the end of the Section "Method" (in addition, the description of the optimization is also changed according to your comment further down).

- There are language errors and typos, e.g., constrain instead of constraint in pg 8 line 162.

- Thank you. We have corrected this and other errors. In the revised manuscript, we have used Wiley Editing Services for proper English language, grammar, punctuation, spelling, and overall style of the manuscript (most of their suggestions).

- Illustrated PSD in Fig. 5 is a little confusing. What are the time step and the time interval for the PSD analysis performed in this figure?

All PSD are generated with the same method. For clarification, the following description is added to the end of section **Method > Spectral Analysis**:

*"The length of the chunks is a compromise between the accuracy of the PSD estimates (smaller chunks, i.e., more chunks) and the frequency resolution and the lowest resolvable frequency (longer chunks). In this study, a length of 256 data points was chosen (10 days and 16 hours), giving 135 overlapping chunks for the two-year long hourly time series. The PSD estimates will therefore be generated for frequencies between $(256\ h)^{-1}$ (thus, including PSD estimates for the 3-4 day period of the time scale of migratory low-pressure systems at mid and high latitudes) and the Nyquist frequency of $(2\ h)^{-1}$ with a resolution of $(256\ h)^{-1}$."*

The wording of the caption and label in Fig. 5 are also changed a little bit

- Could the authors bring more details into the mathematical presentation of the optimization objective function represented by eq.2? The PSD of which function is going to be minimized in the specified frequency range.

  Yes. One additional equation is added, describing the power output time series of wind farm portfolios (Eq. (2) in the re-submitted manuscript). It is the PSD of this equation, which is used to derive an optimized portfolio, being the portfolio where the fluctuations of the total wind power output time series are minimized for frequencies between $(3\ h)^{-1}$ and $(2\ h)^{-1}$. The section describing the optimization has been changed accordingly in the re-submitted manuscript, as an attempt to improve the description.

- Have the authors tested different frequency ranges, and why is the frequency range $(2h)^{-1}\ (3h)^{-1}$ chosen for the optimization?

  The focus is mainly on frequencies between $(3\ h)^{-1}$ and $(2\ h)^{-1}$. Arguments for why these frequencies are considered are added to the: Abstract, Introduction, Method > Spectral Analysis, and Discussion

We did not initially test other frequency ranges. However, we have added a subsection (subsection 4.5 in the revised manuscript).